# A large and diverse autosomal haplotype is associated with sex-linked colour polymorphism in the guppy

Josephine R. Paris [1✉], James R. Whiting[1], Mitchel J. Daniel[2], Joan Ferrer Obiol [3], Paul J. Parsons [1,4], Mijke J. van der Zee[1], Christopher W. Wheat[5], Kimberly A. Hughes [2] & Bonnie A. Fraser[1]

Male colour patterns of the Trinidadian guppy (*Poecilia reticulata*) are typified by extreme variation governed by both natural and sexual selection. Since guppy colour patterns are often inherited faithfully from fathers to sons, it has been hypothesised that many of the colour trait genes must be physically linked to sex determining loci as a 'supergene' on the sex chromosome. Here, we phenotype and genotype four guppy 'Iso-Y lines', where colour was inherited along the patriline for 40 generations. Using an unbiased phenotyping method, we confirm the breeding design was successful in creating four distinct colour patterns. We find that genetic differentiation among the Iso-Y lines is repeatedly associated with a diverse haplotype on an autosome (LG1), not the sex chromosome (LG12). Moreover, the LG1 haplotype exhibits elevated linkage disequilibrium and evidence of sex-specific diversity in the natural source population. We hypothesise that colour pattern polymorphism is driven by Y-autosome epistasis.

[1] Department of Biosciences, University of Exeter, Stocker Road, Exeter EX4 4QD, UK. [2] Department of Biological Science, Florida State University, 319 Stadium Drive, Tallahassee, FL 32304, USA. [3] Departament de Microbiologia, Genètica i Estadística and Institut de Recerca de la Biodiversitat, Universitat de Barcelona, Barcelona, Catalonia, Spain. [4] NERC Environmental Omics Facility, School of Biosciences, University of Sheffield, Sheffield S10 2TN, UK. [5] Department of Zoology, Stockholm University, Stockholm, Sweden. ✉email: j.r.paris@exeter.ac.uk

Understanding the genetic architecture of highly diverse and ecologically-important traits is a fundamental problem in evolutionary biology. This is particularly true for traits where the underlying genes are predicted to be on the Y chromosome. Because the Y-chromosome does not recombine, it is expected to degrade over time, whilst its unique inheritance from fathers to sons will select for genes that increase male fitness[1]. However, this model neglects important factors such as pleiotropy and epistasis[2,3]. Sex-linked colour polymorphism provides a tractable trait for exploring the evolutionary and ecological drivers of balancing selection and sex-chromosome evolution[4,5], and genome sequencing methods have hugely enhanced our ability to detect the genetic basis of colour traits[6]. Using a unique breeding strategy designed to delineate regions of the genome related to colour, we analyse whole-genome sequencing data to uncover the genetic basis of sex-linked colour polymorphism in the Trinidadian guppy (*Poecilia reticulata*).

Guppy colour traits have fascinated biologists for a hundred years, and present an exciting system for testing predictions of sex-linked polymorphic traits. Males display a mosaic of complex and diverse colouration patterns, varying in colour, number, shape, size, and position of spots, while females are a drab and uniform tan colour[7,8]. Guppy colour patterns exhibit high levels of standing genetic variation[9–11], despite evidence that mate choice and predation impose directional selection[12–15]. Considerable evidence suggests that genetic diversity is maintained by negative frequency dependent selection (NFDS), driven by female mate preference for rare or novel morphs[16–20] and also frequency-dependent survival[21,22]. Despite this great diversity in colour patterns, and our understanding of the evolutionary processes maintaining it, the underlying genetic architecture remains largely unknown.

It has long-been hypothesised that colour patterning genes and the sex determining locus (SDL) form a Y-linked 'supergene' in the guppy[23,24]. The supergene-hypothesis originates from early pedigree studies that found patrilineal inheritance for many guppy colour pattern elements[25,26]. While clearly demonstrating the importance of the Y-chromosome for colour, this early work also reported among-population variability in the strength of Y-linkage, indicating a more complex genetic architecture. Indeed, although there is an enrichment of pigment genes on male-specific contigs[27], recent studies have also highlighted the importance of X-linked and autosomal inheritance of colour traits. A QTL mapping study found that only 13% of colour trait loci mapped to the sex chromosome (LG12)[28] and a pedigree analysis of colour pattern inheritance showed that ornaments are not completely Y-linked, hypothesising a potential role for Y-autosome epistasis[29]. Combined, this work suggests that a non-recombining, male-specific region on LG12 plays an important, but not exclusive role in guppy colour patterning.

Recent genomic studies have therefore focussed on identifying the boundaries and genetic content of the non-recombining Y-region. Using male specific diversity across multiple populations, the Y-specific region was concluded to be small, possibly only a single gene, occurring near the distal end of LG12[30–32]. Indeed, sex has been mapped to the distal end of LG12 in multiple mapping and pedigree crossing studies[28,30,33]. Other studies however, conclude that the non-recombining region extends beyond this distal region in some populations based on similar sex-specific genomic diversity measures[34,35]. Intriguingly, the distal end of LG12 has been found to be highly diverse *among* males with many segregating male-specific variants, indicative of multiple Y haplotypes, as would be predicted under NFDS for Y-linked colour traits[31,35]. Within this candidate region, however, no gene associated with colour or sex has been found. Clearly, more work is needed to delineate the Y-linked genomic architecture of this important trait.

In this work, we use an innovative approach to identify genomic regions associated with highly variable, sex-linked, colour polymorphism in guppies. We phenotype and genotype four 'Iso-Y' lines, which originated from a natural population, and show strong Y-linked parental heritability in colour pattern[19,36]. Each Iso-Y line was founded by one male showing a distinct colour pattern. This colour pattern was introgressed onto a randomised genetic background by mating males that resembled the founder, to females from a randomly-mating stock population. Outcrossing to stock population females occurred every 2–3 generations over the 40 generations that the lines were propagated. Given that each backcrossed generation theoretically reduces the parental genome by half, we expect more than 99.99% of the genome to be homogenised through this approach[37]. This experimental design allows us to delineate regions of the genome related to colour pattern, as it should homogenise regions unrelated to the colour differences among lines. Using a Pool-seq approach on each Iso-Y line, we find that the lines are consistently different on an autosome (LG1), not the sex chromosome (LG12). In order to examine the LG1 colour-linked candidates in a natural population, we then conduct an analysis of whole-genome sequencing (WGS) data from the source population, finding that a large and diverse autosomal haplotype is maintained in nature. We hypothesise that colour pattern variation is driven by Y-autosome epistasis. This genetic architecture could help to explain how high levels of Y-linked diversity is maintained in guppy colour patterns.

## Results

**Iso-Y lines are distinct across multiple dimensions of colour.** Male guppies have an ultraviolet component to their colour patterns, and guppies can detect and adjust their social behaviour based on ultraviolet colouration[38]. Consequently, we used multispectral digital photography to capture human-visible and ultraviolet images of males from each Iso-Y line. We used geometric morphometrics to correct for individual differences in body size and shape among fish so that colour patterns could be measured as though they existed on identical male bodies. We then used the Colormesh pipeline[39] to extract colour measurements from these images at sampling locations across the body and caudal fin.

Discriminant analysis of principal components (DAPC)[40,41] (see 'Methods' for details) revealed that males from the different Iso-Y lines were well-separated based on colour pattern. Discriminant functions (DF) 1, 2 and 3 accounted for 51.9%, 36.8%, and 11.4% of the variation among the Iso-Y lines, respectively (Fig. 1a), and distinguished Iso-Y9 from the other Iso-Y lines along axis 1. Axis 2 predominately separated Iso-Y10, and axis 3 subtly differentiated the lines (Supplementary Fig. 1). Colour variation on the caudal peduncle and in the anal region were most important for differentiating the Iso-Y lines (Fig. 1b). Variation in the red colour channel associated with DF1 captured the orange colour spot observed on the anal region in Iso-Y9 (Fig. 1c).

The Iso-Y lines also showed robust phenotypic differences when we examined mean colour measures using permutational MANOVA. Here, we reduced the dimensionality of the colour pattern data using PCA, with 17 PC's explaining 59.3% of the total variation in colour measurements (see Supplementary Fig. 2 for PC distributions of each of the Iso-Y lines). The omnibus test indicated significant overall differences among Iso-Y lines (df = 3169, pseudo-F = 23.66, $P < 0.001$). Post-hoc pairwise tests revealed significant differences in colour pattern among all pairs (all $P < 0.001$; see Supplementary Table 1). Based on centroid distances, the greatest phenotypic differences were between Iso-

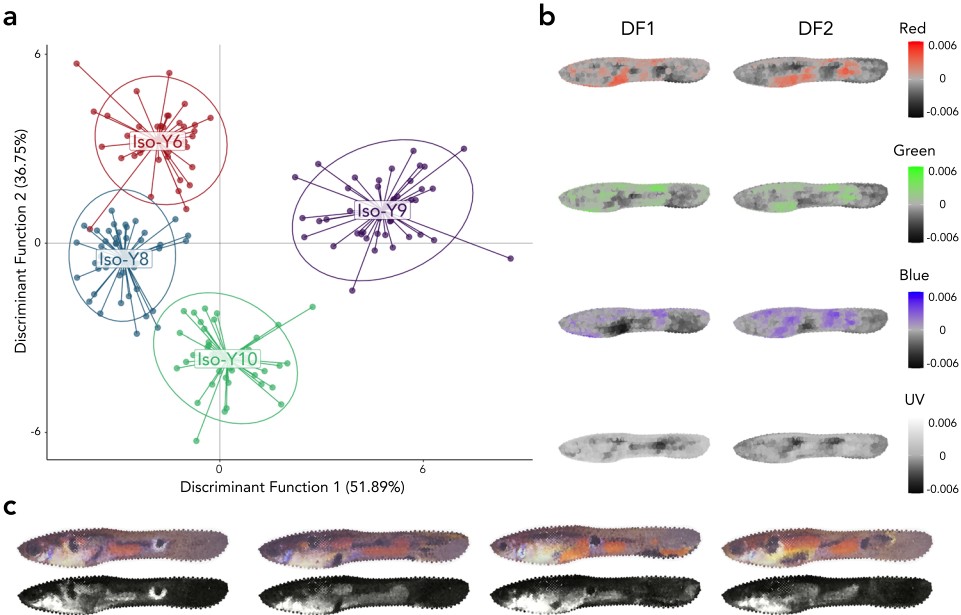

**Fig. 1 Discriminant analysis of principal components (DAPC) differentiating colour measurements among Iso-Y lines. a** Scatterplot of the first 2 Discriminant Functions: Discriminant Function 1 (DF1) and Discriminant Function 2 (DF2). Each point represents a male, and colour denotes the Iso-Y line. **b** Heatmaps for each colour channel depicting the correlation between colour at each sampling location for Discriminant Function 1 (DF1) or Discriminant Function 2 (DF2). **c** Images of the male closest to each Iso-Y line's centroid, constructed from the Red Green Blue (RGB; top) and ultraviolet (UV; bottom) colour measurements at each sampling location. UV images are false colour, with lighter grey indicating higher UV reflectance. Source data underlying Fig. 1a are provided as a Source data file.

Y6 and Iso-Y10 and the smallest phenotypic differences between Iso-Y6 and Iso-Y8. Using permutational t-tests to determine whether the Iso-Y lines differ in phenotypic variance, we found that Iso-Y9 was significantly more variable (Supplementary Table 2; Supplementary Fig. 3).

**Iso-Y lines are consistently different on an autosome.** Using a pooled whole-genome sequencing (Pool-seq) approach and summarising all pairwise comparisons with a PCA, we were able to identify where along the genome the Iso-Y lines were consistently different (Fig. 2). Pool-seq of each of the four Iso-Y lines ($n_{per\ line} = 48$) resulted in a final dataset of 3,995,905 SNPs. Mean pairwise $F_{ST}$ between the Iso-Y lines was high overall ($F_{ST} = 0.091$). We first used pairwise $F_{ST}$ values among the four Iso-Y lines, and then used a multivariate approach by normalising $F_{ST}$ to Z-$F_{ST}$ and summarising these with PCA $F_{ST}$[42]. The aim here was to summarise covariance of $F_{ST}$ among comparisons and assess whether certain regions were consistently differentiated among the Iso-Y lines. Z-$F_{ST}$ PC1 accounted for 37% of the total variance and reflected positive covariance in all pairwise $F_{ST}$ comparisons (Supplementary Table 3).

LG1 and LG12 showed excess divergence among the Iso-Y lines and were clear outliers compared to the rest of the genome (Fig. 2a). LG1 and LG12 both had the highest chromosome-wide average Z-$F_{ST}$ PC1 scores (LG1 = 1.48; LG12 = 1.62) and the highest percentage of SNPs with a Z-$F_{ST}$ PC1 score above an upper 95% quantile of 3 (LG1 = 24%; LG12 = 29%). The remaining 47% of high-scoring SNPs were distributed across the genome, with other individual chromosomes or scaffolds accounting for <7% of high-scoring SNPs (according to an upper 95% quantile of 3; Supplementary Fig. 4). Other chromosomes showed inconsistent regions of localised differentiation (Supplementary Fig. 5) and we found no evidence indicative of a genome-wide relationship between recombination and Z-$F_{ST}$ PC1 (Supplementary Fig. 6). Thus, LG1 and LG12 became our focus for investigating differentiation among the Iso-Y lines.

Interestingly, the patterns of differentiation were different for these two focal chromosomes. Z-$F_{ST}$ PC1 indicated three regions of high differentiation on LG1 (Fig. 2b). On LG12, these scores were elevated consistently along the entire chromosome (Fig. 2d).

By performing an additional PCA on LG1, we found good agreement between the areas of differentiation amongst the Iso-Y lines. Z-$F_{ST}$-LG1 PC1 (52% of the total variance) showed high positive loadings among five of the six pairwise comparisons (Iso-Y6–Iso-Y8 being the exception; Supplementary Table 4), which was also reflected in the per-SNP pairwise $F_{ST}$ comparisons, where Iso-Y6 and Iso-Y8 exhibited the lowest differentiation (Supplementary Fig. 7; Supplementary Table 5). The Iso-Y6–Iso-Y8 comparison loaded positively onto PC2 (17% of total variance) and Z-$F_{ST}$ PC2 (and to some extent, PC3) highlighted the same regions of differentiation as PC1 (Supplementary Fig. 8). We found no relationship between Z-$F_{ST}$ PC1 scores and the recombination landscape on LG1 (Supplementary Fig. 9).

In contrast, PC axis loadings for Z-$F_{ST}$ on LG12 were different among the Iso-Y lines (Supplementary Table 6). This was also apparent in per-SNP pairwise $F_{ST}$ values (Supplementary Fig. 10). All comparisons with Iso-Y9 loaded strongly onto PC1 (PC1 captured 37% of the total variance), suggesting Iso-Y9 is the most differentiated line on LG12. This was consistent with pairwise $F_{ST}$, where Iso-Y9 had the highest mean pairwise $F_{ST}$ SNPs (>0.2) and thus high Z-$F_{ST}$-LG12 PC1 scores reflected the haplotype associated with the Iso-Y9 phenotype. Z-$F_{ST}$-LG12 PC2 (PC2 captured 30% of total variance) loadings were high for the remaining two comparisons with Iso-Y6. This suggests SNPs with high Z-$F_{ST}$-LG12 PC2 scores reflect the haplotype associated with the Iso-Y6 phenotype. Z-$F_{ST}$-LG12 PC3 reflected the final comparison, Iso-Y8-Iso-Y10. Z-$F_{ST}$-LG12 PC2 and PC3 also reflected Iso-Y line-specific loadings (Supplementary Fig. 11). Taken together, this demonstrates that Iso-Y specific haplotypes co-occur at the same regions on LG1, whereas on LG12, the Iso-Y specific haplotypes occur in different regions.

We were able to further identify areas of consistent divergence in three regions on LG1 using change point detection (CPD) on

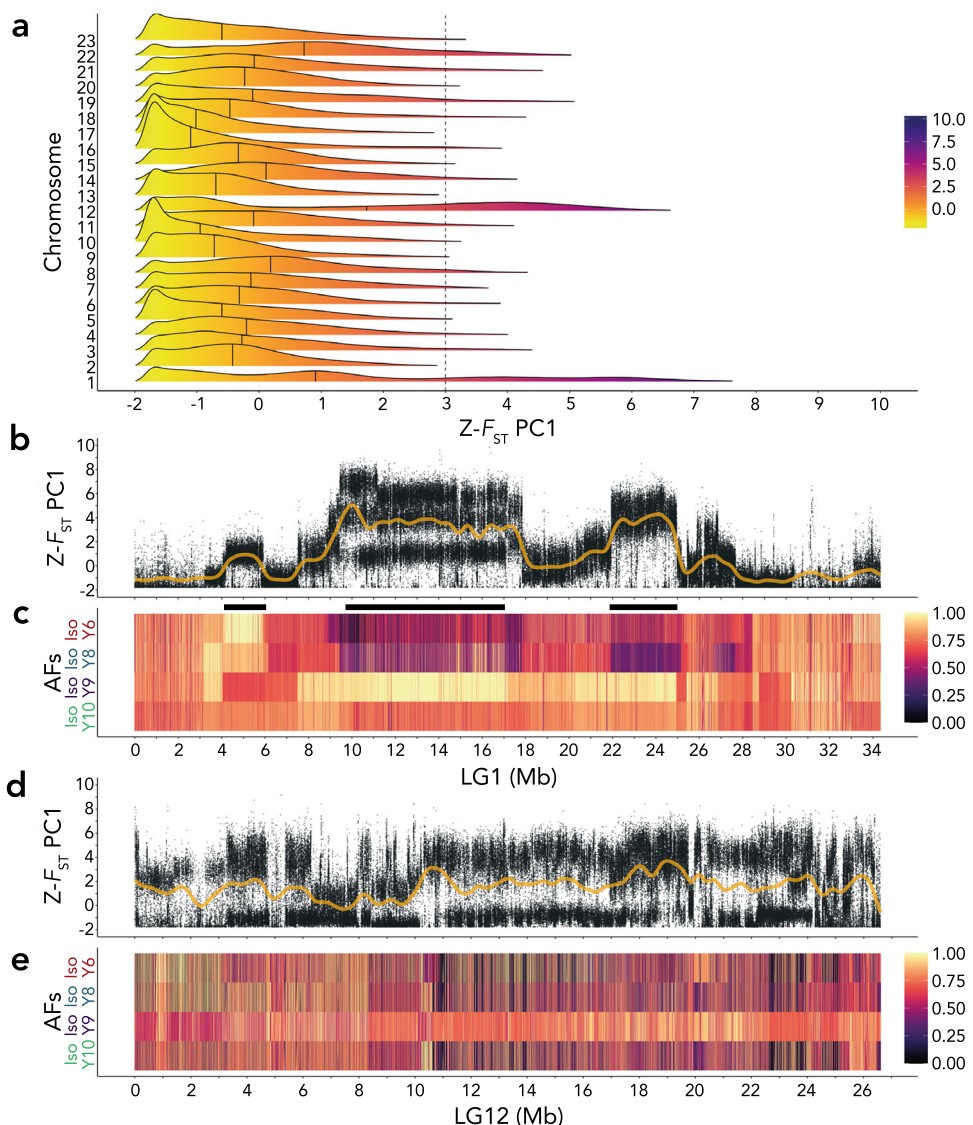

**Fig. 2 Genetic differentiation between the four Iso-Y lines: Iso-Y6; Iso-Y8; Iso-Y9; Iso-Y10. a** Density plots of the SNP distribution of Z-$F_{ST}$ PC1 $F_{ST}$ for each chromosome of the guppy genome (23 chromosomes). The colour scale represents the Z-$F_{ST}$ scores as depicted in the legend (yellow: low-scoring SNPs; indigo: high-scoring SNPs). Black lines within each density curve mark the median value of each chromosome. Dashed *x*-axis intersect marks the upper 95% quantile of 3. **b** Per-SNP Z-$F_{ST}$ PC1 scores for LG1; yellow line represents a smoothed spline of the data. **c** Allele frequency (AF) plots of LG1 for each Iso-Y line (Iso-Y6—red; Iso-Y8—blue; Iso-Y9—purple; Iso-Y10—green). AFs were polarised to the major allele of Iso-Y9. Changes in the AFs are represented by the colours depicted in the legend. On LG1, Change Point Detection (CPD) pinpointed three regions of consistent differentiation, which are indicated at the top by black rectangles: Region 1 (4–5.9 Mb); Region 2 (9.6–17 Mb); Region 3 (21.9–24.9 Mb). **d** Per-SNP Z-$F_{ST}$ PC1 scores for LG12; yellow line represents a smoothed spline of the data. **e** Allele frequency (AF) plots of LG12 for each Iso-Y line. AFs were polarised to the major allele of Iso-Y9 and changes in the AFs are represented by the colours depicted in the legend. On LG12, Change Point Detection (CPD) did not uncover any consistent regions of differentiation between the Iso-Y lines. X ticks are displayed in Megabases (Mb). Source data underlying all components of Fig. 2 are provided as Source data files.

the allele frequencies (AFs—polarised to the major allele of Iso-Y9) and Z-$F_{ST}$ PC1 (Fig. 2c). Region 1 encompassed ~1.9 Mb (coordinates: 4,079,988–5,984,584 bp). Region 2 encompassed ~7.4 Mb (coordinates: 9,627,619–17,074,870 bp). Region 3 encompassed ~3 Mb (coordinates: 21,944,840–24,959,750 bp). Using the same CPD methods, no regions were consistently differentiated between the Iso-Y lines on LG12 (Fig. 2e).

Next, to explore diversity within the Iso-Y lines, we calculated π across 10 kb windows for our two focal chromosomes (LG1 and LG12) and performed the same multivariate approach as above. On LG1, we found that regions of differentiation could be explained by a shared chromosomal landscape of diversity among all Iso-Y lines with Z-π-LG1 PC1 (PC1 captured 74% of the

variance) accounting for the majority of variance in diversity (Supplementary Fig. 12; Supplementary Table 7). We found no evidence of increased coverage associated with increased diversity in the regions of LG1 (Supplementary Fig. 13). On LG12, Z-π PC1 accounted for most of the variance (PC1 captured 88% of the variance) and showed a significant (>99% upper quantile of 4.6) peak in diversity at 24.27 Mb (Supplementary Fig. 14; Supplementary Table 8). Z-π-LG12 PC2 (PC2 captured 7% of the variance) showed a high level of residual variance in π, associated with the variance of Iso-Y9, that was not otherwise explained by the general chromosomal landscape. The diversity hotspot (peak at 24.28 Mb, >99% upper quantile of −1.4) represented by PC2 is thus unique diversity within Iso-Y9. Both LG12 peaks: 24.27 Mb

($\pi$ shared by all Iso-Y lines); and 24.28 Mb ($\pi$ unique to Iso-Y9) overlap with previously recorded high male diversity at the putative non-recombining Y[31].

**Evidence of multiple haplotypes and candidate colour genes on LG1.** In Region 2 (9.6–17 Mb), two bands of $F_{ST}$ were apparent in analysis of Z-$F_{ST}$-LG1 PC1, and in pairwise $F_{ST}$ (Fig. 2b; Supplementary Fig. 7). To explore this further, we assessed the segregation of the AFs within each of the three identified regions in more detail (Supplementary Fig. 15). Corresponding to the double-banding of $F_{ST}$, Region 2 showed unusual AF patterns within the lines. Iso-Y9 showed fixation, but the other three Iso-Y lines showed multiple bands of AFs, which taken together did not sum to 1. Additionally, in Region 3, Iso-Y6 also showed two sets of distinct bands of AFs. Assessment of the AF density distributions showed clear patterns of bimodality (trimodality in Iso-Y6) in Region 2 (Supplementary Fig. 16) and bimodality in Iso-Y6 in Region 3 (Supplementary Fig. 17). The distinct patterning corresponding to the different bands of AFs could suggest strong linkage between the SNPs segregating in each band, indicative of multiple maintained haplotypes. We reasoned that a complex haplotype structure exists, in which alleles associated with more recently derived haplotypes are nested within an older haplotype (Fig. 3). Bimodal AF distributions parsimoniously represent AFs associated with the older (larger density peak) and younger, derived (smaller density peak) haplotypes.

We found several strong candidates for colour, male-specific fitness and vision in the three differentiated regions on LG1 (see Supplementary Data 1 for a full list of gene annotations). Region 1 (4–5.9 Mb) contained 62 predicted genes. Of interest was *tll1*, with a role in caudal fin, dorsoventral patterning in *D. rerio*[43], *pcm1* involved in spermatogenesis[44], and *ptpn13* which is Y-linked in humans[45] and in some fish due to its physical linkage with *gsdf*, a gene which is highly conserved in fish sex differentiation pathways[46,47]. Region 2 (9.6–17 Mb) contained 291 predicted genes. Genes with a potential role in colour included *xpa*, involved in pigmentation and photosensitivity to UV light[48], *pcdh10a* involved in melanocyte migration, *crebbpa*, which has been identified as a candidate for plumage colouration in chickens[49], and *shoc2*, which causes pigmentation abnormalities[50], as well as five keratin genes, which have a role in pigmentation[51]. We also identified five retinal genes (*slc24a2*[52], *stra6l*[53], *pnpla6*[54], *cabp2a*[55] and *nxnl1*[56]), and three genes involved in spermatogenesis or sperm motility (*tdrd7a*[57], *nanos3*[58] and *tekt4*[59]). Region 3 (21.9–24.9 Mb) contained 94 predicted genes. This region also contained several promising candidates including two paralogs annotated as *kita*, previously identified as a key gene involved in pigment pattern formation in guppy strains[60] and zebrafish[61], *sox10a*, a sex determining region Y-box that regulates the expression of the *mitf* gene, which is the master regulator of melanophore–melanocyte differentiation in teleosts[62], and is also responsible for colouration in rock pigeons[63], *mchr1*, a melanin-concentrating hormone receptor, and a *TRYP*, located in melanocytes and involved in the production of melanin[64]. We also performed an assessment of Gene Ontology (GO) enrichment and KEGG mapping for the three regions on LG1 (see Supplementary Data 2 for results). Analysis using gene annotation information combined across all three LG1 regions showed a significant KEGG mapping to Lysosome (11 genes). This is of potential interest given that the melanosome is a lysosome-related organelle.

**Evidence for the LG1 haplotype in the natural population.** We next examined natural data by performing whole-genome sequencing of 26 wild-caught guppies from the source

population ($n_{females} = 16$, $n_{males} = 10$, average coverage ≥13×, final dataset = 1,021,495 SNPs). Using multiple lines of evidence, we found a large haplotype on LG1 (11.1–15.9 Mb) segregating in the natural source population, which lies within 'Region 2' identified in the Iso-Y analysis (9.6–17 Mb). We term this area 'Region 2-NP' (Region 2 Natural Population) (Fig. 4).

Analysis of linkage disequilibrium (LD) on LG1 revealed an area of high linkage between 11,114,772 and 15,890,374 bp, where it was apparent that several overlapping linkage blocks exist (Fig. 4a). We also analysed patterns of LD for males and females separately and found that males exhibited at least two distinct neighbouring linkage blocks, but females showed high linkage across the entire region (Supplementary Fig. 18). We then assessed shifts in local ancestry by performing a local PCA approach[65], which recapitulated the identified areas of high LD (Fig. 4b), with a pattern of significant differentiation (saturated eigenvalues > 0.01) represented by three overlapping multidimensional scales (MDS) on LG1: MDS1: 11,216,906–15,375,083 bp; MDS2: 11,430,012–14,570,259 bp; MDS3: 12,326,328–15,308,055 bp. This demonstrates that subsets of correlated SNPs within Region 2-NP exhibit ancestry relationships that deviate from those observed across the rest of the chromosome; a pattern indicative of inversions, long haplotypes under balancing selection, reduced recombination, or changes in gene density[65].

Intersex $F_{ST}$ showed differences between the sexes within Region 2-NP (Fig. 4c); in particular, an area of high density of elevated intersex $F_{ST}$ between 12.2 Mb to 13.1 Mb. We found that elevated intersex $F_{ST}$ was driven by a reduction in male-specific diversity, as measured by intersex $D_a$ ($D_{XY}$ - female $\pi$; Fig. 4d). Overall, these results suggest that selection is operating differently between the sexes in the candidate region.

As recombination history is known to affect the maintenance of tightly linked genetic architectures[66], we also examined the genome features of Region 2-NP. We did not observe any differences in the proportion of GC content (Supplementary Fig. 19), nor repeat elements (Supplementary Fig. 20) in our candidate region. SNP density within Region 2-NP was considerably higher compared to the rest of the chromosome (Supplementary Fig. 21), but there was no evidence of extremes in read depth indicative of duplication or copy-number variation (Supplementary Fig. 22). Gene density showed a moderate number of genes across Region 2-NP, and a drop in gene density at ~13.2 Mb (Supplementary Fig. 23).

We performed similar analyses on LG12, confirming previously identified differences between the sexes[30,31,35,67]. LD analysis showed two main regions of increased linkage; a fragmented region extending from 4.6 to 6.7 Mb, and another more clearly defined region near the terminal end of the chromosome (23.8–25.3 Mb) (Fig. 4e). A local PCA of LG12 identified two significant MDS axes (saturated eigenvalues >0.01) approximately corresponding with the areas of high linkage (Fig. 4f). MDS1 outliers were observed at the terminal end of the chromosome (coordinates: 21,913,633–25,664,533 bp). MDS2 depicted an area encompassing the latter part of the high linkage block, extending past it in the latter coordinates (coordinates: 5,608,703–7,063,435 bp; Fig. 4e).

Overlapping with MDS2, a significant elevation in intersex $F_{ST}$ was observed (Fig. 4g) and in the latter part of the region, a narrow peak in significantly increased male diversity was detected at 5.67 Mb (Fig. 4h). This second region overlaps with a previously identified $D_a$ outlier window in an analysis of six natural populations (6.94–6.95 Mb and 7.00–7.01 Mb), where male-specific $D_a$ was notably higher in high-predation (HP), compared to low-predation (LP) populations[31]. Moreover, the LD in this region extends across 4.6–6 Mb which includes another previously identified male-biased candidate region (LG12:

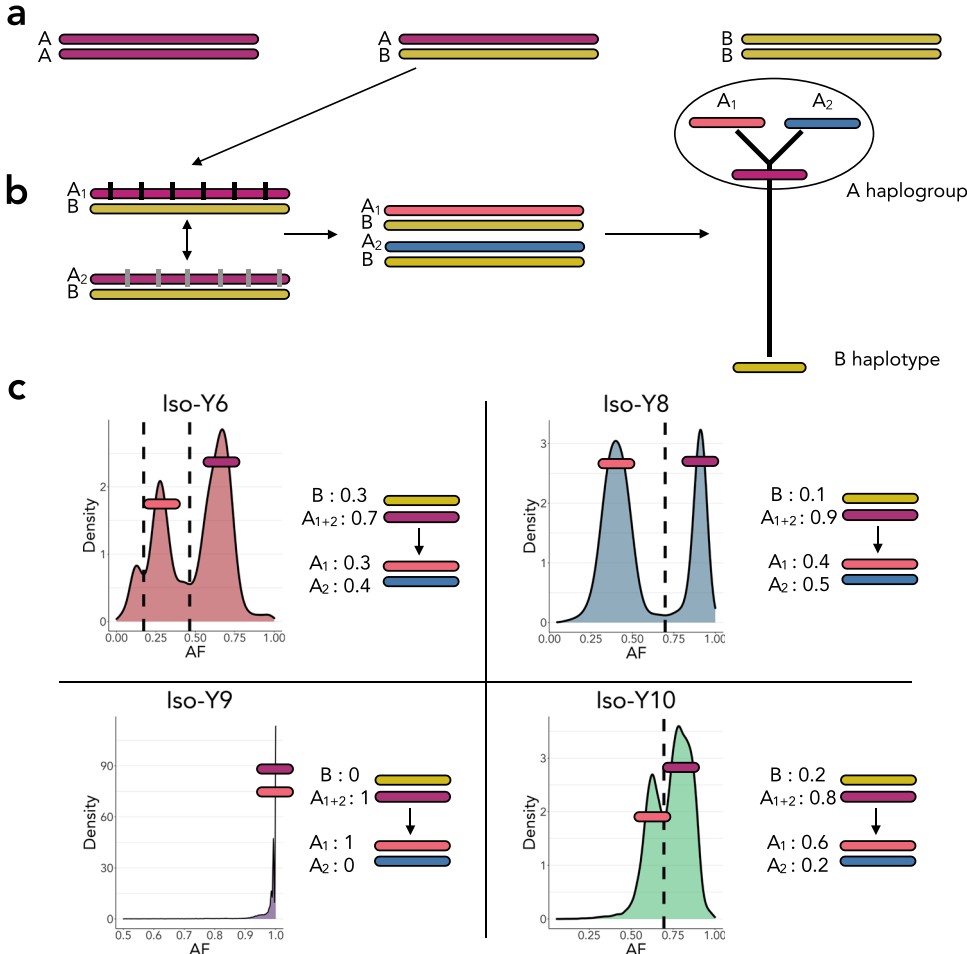

**Fig. 3 Schematic representation of the multiple bands of allele frequencies present in the Iso-Y Pool-seq data. a** AA represents homozygous, AB represents heterozygous, and BB represents homozygous alternative. **b** The ancestral A haplotype accumulates SNPs, differentiating it into two versions of the A haplotype: A1 and A2. The 'Y' shaped haplotype tree represents the predicted evolutionary relationships between the 'A haplogroup' composed of ancestral A, derived A1 and A2 haplotypes, and the 'B haplotype'. Branch lengths represent evolutionary distance, and thus SNP count. We predict that Iso-Y9 is fixed for one of the derived A haplotypes. For illustration purposes, we have shown fixed allele frequencies (AFs) for the A1 haplotype. The remaining Iso-Y lines are all heterozygous with the AB genotype. There are no BB individuals. Individuals within the Iso-Y6, Iso-Y8 and Iso-Y10 pools have segregating A1 and A2 haplotypes, which when compared to Iso-Y9 A1 show multiple bands of AFs, as shown in the AF calculations in part (**c**). Due to the nature of Pool-seq, it is unclear what the actual genotypes are. We focus on providing an explanation of the bimodal peaks, but it is noteworthy that in Iso-Y9 there are many fixed sites, but also some 'nearly fixed' sites, which suggests some diversity also exists in the Iso-Y A1/A1 haplotype, which likely represents the trimodality of Iso-Y6, (i.e. further complexity in the A1/A1 haplotype that's captured in comparisons with Iso-Y6). Refer to Supplementary Figs. 16 and 17 for further visualisation of the segregating AFs.

4.8–5.2 Mb)[31]. An assessment of the gene annotations within the MDS2 coordinates identified only a few candidates of interest (Supplementary Data 3): *dmgdh*, a gene that affects sperm trait variation and is part of a sex-supergene in songbirds[68] and *bhmt*, a folate-related gene that shows an association with skin pigmentation in humans[69].

It is predicted that the terminal region of LG12 contains the sex-determining locus (SDL)[28,70,71] and a recent multiple population genomics survey identified a sex-linked region between 24.5 and 25.4 Mb in LG12, which overlaps with the region of high LD and MDS1 region identified here[31]. Moreover, analysis of intersex $F_{ST}$ showed high differentiation between the sexes in the region (Fig. 4g), and analysis of intersex $D_a$ revealed that this was driven by an excess of male diversity (Fig. 4h). In contrast to LG1, which showed that elevated intersex $F_{ST}$ was driven by reduced male diversity, the high diversity observed here is consistent with a hypothesis of multiple diverse Y-haplotypes and NFDS on the Y[35]. Previous investigation of the gene content within the terminal region found it to be relatively gene-poor,

containing multiple repeated copies of NLRP1-related genes[31], which form part of the inflammasome[72] and have no known role in sex determination. Our analysis of gene content for the slightly expanded region found in this population identified a single potential colour pattern candidate (*vldlr*), which is responsible for caudal fin patterning in *Danio rerio*[73], and two male-trait candidates: *AIG-1* (androgen-induced protein-1) family, and *spag16* (sperm-associated antigen 16) (Supplementary Data 3).

**Pinpointing informative breakpoints within the LG1 haplotype.** Combining both the natural population data and our experimental Iso-Y lines, we found evidence for the maintenance of a large, highly diverse haplotype on LG1 associated with male-specific colour. In the Iso-Y lines, Region 2 encompassed multiple bands of tightly associated allele frequencies, suggesting multiple derived haplotypes. Analysis of the natural population data revealed that Region 2-NP is characterised by elevated linkage, and is defined by linkage patterns indicative of not just one, but multiple derived local ancestries. Comparisons of male and

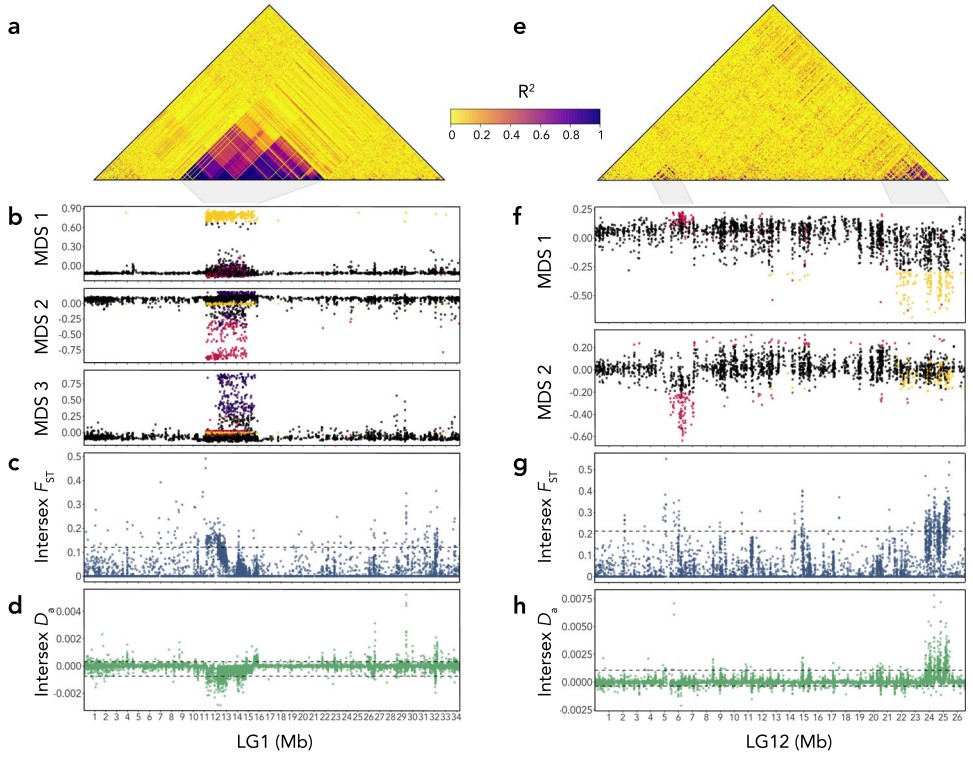

**Fig. 4 Analysis of LG1 and LG12 in the natural source population ($n_{females} = 16$, $n_{males} = 10$). a** LG1 heatmap of patterns of linkage disequilibrium (LD) measured as $R^2$. Amount of LD is shown by colour intensity as depicted in the legend (yellow: low LD; indigo: high LD). High LD is observed at coordinates: 11,114,772–15,890,374). **b** LG1 local PCA in 10 bp windows, depicting three significant multidimensional scales (MDS): MDS1 (yellow, coordinates: 11,216,906–15,375,083 bp); MDS2 (pink, coordinates: 11,430,012– 14,570,259 bp); MDS3 (purple, coordinates: 12,326,328–15,308,055 bp). **c** Intersex $F_{ST}$ calculated in 1 kb windows across LG1. Dashed line marks the 95% quantiles. **d** Intersex $D_a$ calculated in 1 kb windows across LG1. Dashed lines mark the 5% and 95% quantiles. **e** LG12 heatmap of patterns of linkage disequilibrium (LD) measured as $R^2$. Amount of LD is shown by colour intensity as depicted in the legend (yellow: low LD; indigo: high LD). High LD is observed between ~4.6–6 Mb and at the terminal region between ~23.8–25.3 Mb. **f** LG12 local PCA in 10 bp windows, depicting two significant multidimensional scales (MDS): MDS1 (yellow, coordinates: 21,913,633–25,664,533 bp); MDS2 (pink, coordinates: 5,608,703–7,063,435 bp). **g** Intersex $F_{ST}$ calculated in 1 kb windows across LG12. Dashed line marks the 95% quantile. **h** Intersex $D_a$ calculated in 1 kb windows across LG12. Dashed lines mark the 5% and 95% quantiles. X ticks are displayed in Megabases (Mb). Source data underlying all components of Fig. 4 are provided as Source data files.

female diversity within the population suggests selection may operate differently between the sexes in Region 2-NP; in particular, between 12.2 Mb and 13.1 Mb, which showed high intersex $F_{ST}$ and a reduction in male diversity. Structural variant (SV) analyses using short-read and long-read data did not show support for SVs in Region 2 nor Region 2-NP (Supplementary Data 4). Our prediction is that the region encompasses a large and diverse haplotype, and that within this haplotype, genetic rearrangements have resulted in several derived haplotype segments. To interrogate this further, we performed an in-depth analysis.

Individual genotypes of the Iso-Y lines and the natural population data across LG1 revealed long stretches of maintained genotype states within Region 2-NP (Fig. 5a). In the Iso-Y line data, Iso-Y9 showed distinct blocks of fixed HOM ALT genotypes, whilst the other Iso-Y line genotypes were entirely heterozygous (HET). This observation is congruous with the multiple bands of allele frequencies observed in the Iso-Y line data. Of individuals from the natural population ($n_{total} = 26$), 17 ($n_{females} = 10$, $n_{males} = 7$) shared the same overall genotype signature of HOM ALT genotype blocks as seen in Iso-Y9. Seven individuals ($n_{females} = 4$, $n_{males} = 3$) showed genotype blocks alternating between areas with the Iso-Y9 HOM ALT genotype signature, and areas of heterozygosity. Lastly, only two individuals showed a signature of extended blocks of HOM REF genotypes:

NAT08 and NAT16 ($n_{females} = 2$, $n_{males} = 0$), yet HOM REF genotypes were only maintained for a proportion of the entire region.

By phasing the heterozygous individuals ($n_{total} = 9$), we found evidence of two large divergent haplotypes encompassing the entirety of Region 2-NP with subsequent recombination at conserved breakpoints (Fig. 5b). This indicates that the whole high linkage block is not one large inversion. We identified six repeated breakpoints (i.e. occurring in; ≥2 individuals): BP1: 11.6 Mb ($n_{females} = 2$); BP2: 11.7 Mb ($n_{females} = 3$); BP3: 12.2 Mb ($n_{females} = 3$, $n_{males} = 1$); BP4: 13.1 Mb ($n_{females} = 2$, $n_{males} = 2$); BP5: 14.4 Mb ($n_{females} = 2$); BP6: 15.4 Mb ($n_{females} = 2$).

These analyses allowed us to pinpoint breakpoints contributing to patterns of differential selection and linkage between the sexes. From the start of Region 2-NP to BP3 (12.2 Mb), there was a distinct difference in zygosity between males and females (Females: 11 HOM ALT, 4 HET, 1 HOM REF; Males: 10 HOM ALT). Therefore, males are out of HWE; based on the female AFs, we would expect 5 HOM ALT, 3 HET and 2 HOM REF in males. The region delineated by BP3 to BP4 overlaps with a high density of SNPs showing signatures indicative of sex-differential selection (12.2 Mb to 13.1 Mb). The only breakpoint consistent between males and females (BP4) also distinguishes the break in LD in males, the drop in gene density, and a break in derived ancestry estimates of 100 bp windows (Supplementary Figs. 18, 23, 24).

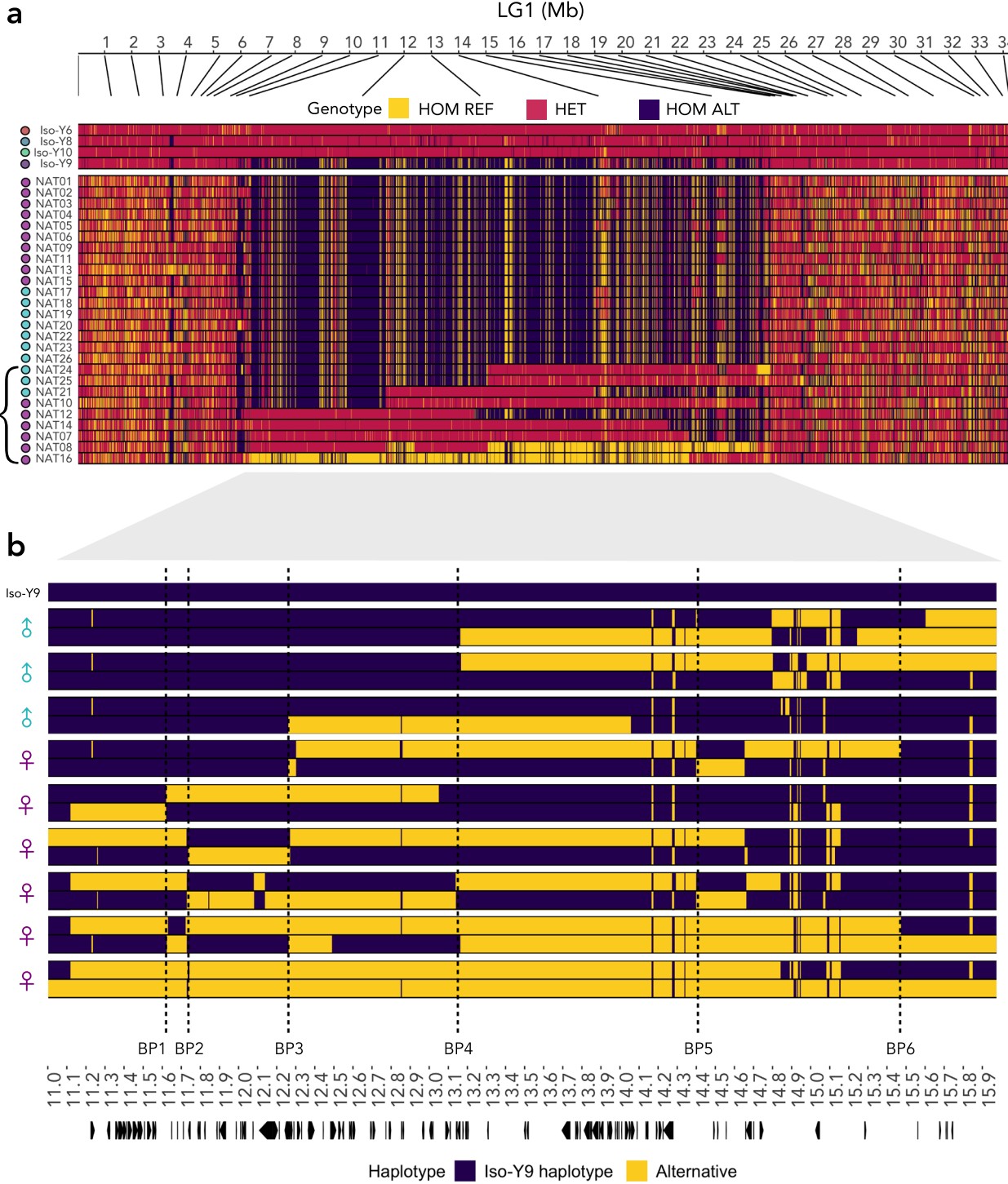

**Fig. 5 LG1 Region 2 haplotype structure. a** Genotype plot for Iso-Y lines and natural data combined showing the genotype of each SNP depicted as homozygous reference (HOM REF: yellow), heterozygous (HET: pink) or homozygous alternative (HOM ALT: purple); each individual is coloured by Iso-Y line, or by sex (females in purple, males in cyan). The bracket shows the heterozygous individuals used in the haplotype analysis in panel (**b**). **b** Haplotype plot of phased data for natural-derived heterozygous samples ($n_{total} = 9$, $n_{males} = 3$, $n_{females} = 6$) polarised to the Iso-Y9 haplotype (purple) and alternative haplotype (yellow). Symbols next to each of the individuals represent sex (females in purple, males in cyan). Breakpoints (BP) in the haplotype are identified when phases switch between purple and yellow. Dashed lines mark conserved BPs ($\geq 2$ individuals): BP1: 11.6 Mb ($n_{females} = 2$); BP2: 11.7 Mb ($n_{females} = 3$); BP3: 12.2 Mb ($n_{females} = 3$, $n_{males} = 1$); BP4: 13.1 Mb ($n_{females} = 2$, $n_{males} = 2$); BP5: 14.4 Mb ($n_{females} = 2$); BP6: 15.4 Mb ($n_{females} = 2$). Arrows at the bottom highlight the location of gene annotations for the region. Source data underlying all components of Fig. 5 are provided as Source data files.

Taken together, this localises our candidate region from the beginning of Region 2-NP to BP4 as both selected in males in a natural population and associated with colour in our breeding design.

## Discussion

Overall, our results reveal a surprising genetic architecture for colour pattern diversity in guppies. Even though the differences in colour pattern between the Iso-Y lines are Y-linked (i.e. inherited faithfully from father to son), consistent genetic differences are most pronounced on an autosome, LG1. Analysis of the phenotype data showed that the breeding design was successful in producing four distinct Iso-Y lines with marked differences in colour pattern. Using these unique Iso-Y lines, we delineated three regions on LG1 that were consistently different among the differently coloured lines, and within these regions we identified many genes known to be linked to colour in model species. Then, using extensive WGS from the source population, we further highlight just one of these regions (Region 2-NP: 11.1–15.9 Mb), as having strong linkage and significant local ancestry, finding that a large and variable haplotype is maintained in nature.

We had hypothesised that the Y-linked colour pattern genes in our Iso-Y lines would be located on the sex chromosome, LG12. If colour pattern traits are fully Y-linked, and each Iso-Y line's unique Y haplotype is fully inherited through the patriline, then with this breeding design we would predict a consistent pattern of high differentiation in the Y-linked region on LG12 between all Iso-Y line comparisons. Instead, we see that overall, the chromosome shows moderate divergence, driven by elevated differentiation in comparisons to Iso-Y9 and localised regions of differentiation in comparisons with Iso-Y6. Whilst elevated $F_{ST}$ across the chromosome is expected due to sex linkage, we found no evidence of a consistently differentiated region on LG12. Moreover, all guppy chromosomes are acrocentric and show evidence of male heterochiasmy[30,67,74]; yet recombination events are particularly rare on LG12, implying that LG12 may decompose at a slower rate compared to autosomes. Our LG1 haplotype on the other hand, shows consistent differentiation between the lines. The near fixation of the LG1 Region-2 haplotype between the lines is strong evidence for this region harbouring genes involved in colour.

An alternative explanation for the association between LG1 Region-2 haplotype and colour is unusually low recombination on LG1 and insufficient backcrossing, resulting in a spurious relationship. However, we think this scenario is unlikely; there is no evidence of unusually low recombination on LG1[30,33] (Supplementary Fig. 9), and as each generation reduces the proportion of parental genome by half, just <0.01% of the parental genome should remain after 13–20 generations of backcrossing. Moreover, we found that Iso-Y6 and Iso-Y8 are the least phenotypically divergent lines, and they also showed the lowest genetic differentiation on LG1.

Our hypothesis is that colour pattern is controlled by the epistatic interaction between Y-specific regions and our candidate region on LG1 in this population. Y-autosome epistasis has been reported previously in other systems. For example, polymorphisms on the gene-poor Y chromosome in *Drosophila* spp. have been shown to differentially affect the expression of hundreds of X-linked and autosomal genes, specifically those that are highly expressed in males, and with clear fitness-related functions in males (e.g. spermatogenesis and pheromone detection[75,76]). Previous research on this population of guppies (Paria) also supports our working hypothesis. Testosterone-treated females from Paria displayed high amounts of colour, suggesting an

autosomal component to colour expression[77]. However, it has also been found that males in Paria exhibit strong patrilineal inheritance of colour pattern, suggesting the importance of the Y[36]. Given the success of our breeding design, where introgressed Iso-Y lines were significantly different in colour, we also found strong evidence for Y-linkage for colour. Ongoing work is directly testing the roles of Y-autosome epistasis versus Mendelian inheritance on guppy colour pattern by conducting crosses between Iso-Y lines with different LG1 haplotypes. Furthermore, it would be interesting to quantify the colour patterns of testosterone-treated females derived from these lines to more fully explore inheritance patterns in females.

By further exploring our natural source population data, we can begin to hypothesise how this epistatic interaction might operate and how it may lead to the maintenance of Y-linked variation in colour. We recorded differences between the sexes in LG1 Region 2-NP, although this region did not appear in a previous analysis examining consistent differences between the sexes across multiple populations[31]. We found that this difference is caused by a reduction in expected diversity in males, where males were not in HWE in the region from 11.1 Mb to 13.1 Mb (start of Region 2-NP to BP4), although females were in HWE. This could be indicative of a deleterious Y-autosome interaction, where certain Y haplotypes are incompatible with the 'REF' LG1 haplotype. Guppy adult sex-ratios, are indeed, generally female-biased[78–81], and show a stronger female-bias in upstream low-predation (LP) environments, like Paria, albeit with sampling variability[78,81]. We found no evidence for large structural rearrangements (such as inversions) underlying the maintenance of our candidate region, LG1 Region 2-NP, and did not detect any relationship with our predicted conserved recombination points, nor increased GC or TE content. Therefore, exactly which molecular traits are driving this haplotype structure is unclear. By examining phased heterozygous individuals, it is however, apparent that crossing-over is restricted to key points along the haplotype and overall linkage is maintained in the population. Models of balancing selection show that balanced variants produce diversity patterns similar to those caused by positive selection[82], and simulations suggest that balancing selection alone can maintain high-LD, and in particular, high divergence between colour phenotypes[83]. Taken together, our natural source population data reveals interesting signals in this region but determining the selective forces responsible for its maintenance requires a larger sample size, and colour pattern phenotype data from the natural source population, particularly for recombinant males.

It has been argued that selection must be extremely strong, or unrealistic, for differences in allele frequencies to be maintained between the sexes on an autosome[84,85]. On the other hand, it is also contended that sex differences on autosomes are artefacts caused by duplications or translocations onto the Y chromosome[86]. Read coverage was not increased in the LG1 region, and we found no evidence of reduced mapping quality, which would be expected if this region was the result of a duplication or translocation event. Nor do we find evidence that our candidate region on LG1 is misassembled; we previously found strong Hi-C contact across the chromosome, although the genome assembly is derived from an individual from a different population[31]. We also found no evidence that a translocation from LG1 to LG12 had occurred uniquely in this population in our SV analyses, and we did not observe elevated inter-chromosomal LD in the natural data (Supplementary Fig. 25). Moreover, linkage along chromosomes, including LG1, has been observed in several different populations and mapping crosses suggesting a translocation is unlikely to be the cause[30,33,67]. We

do however recognise that evolution of sex determination mechanisms and sex chromosomes have occurred in laboratory-adapted populations in a similar number of generations, in both *Danio rerio*[87] and *Xiphophorus*[88].

Early research found that the linkage between colour pattern traits and the Y was under selection, with increased Y-linkage for colour traits in downstream, high-predation (HP) environments and X-linkage in the upstream, low-predation (LP) environments. Whether Y-linkage varies by predation environment has been indirectly studied across Trinidad, where females treated with testosterone exhibited more colour patterns in LP populations compared to their HP counterparts[77,89]. Such observations are also consistent with an autosomal genetic component of colour pattern traits that are under weakened selection in LP environments[90]. Other studies suggest that the size of different strata on LG12 vary between HP and LP, with increased Y-linkage in LP[34] (see also[35,91]), but these results were not repeated across other HP-LP pairs[30–32]. Our candidate region on LG1 may explain differences in Y-linkage between predation ecotypes. Specifically, we identified a region between 12.2 Mb to 13.1 Mb with reduced diversity in males, compared to females, which also corresponded with a break in male patterns of LD. Previous analysis of molecular convergence between HP and LP populations identified two outlier windows in this area (12.17–12.18 Mb and 12.21–12.2 Mb), indicating that this particular region may be under divergent selection for HP-LP phenotypes[92]. We further compared WGS data from other populations across Trinidad at LG1, and detected a strong signal of HP-LP association within 12.2 Mb to 13.1 Mb, but the large haplotype structure is unique to the population studied here (Supplementary Fig. 26). This suggests that this region may be involved in the differential selection of colour phenotypes depending on predation regime.

Based on our Iso-Y lines and natural population data, we hypothesise a Y-autosome epistatic genetic architecture for guppy colour traits. This architecture may be particularly well-suited to a Y-linked trait under NFDS, such as guppy colour pattern. Models suggest that sex-linked polymorphism can only be maintained by natural selection in unusual genetic systems, where the maintenance of Y-linked variation in Y-autonomous models involves frequency-dependent selection, or interactions with other chromosomes[93]. Recognition of the importance of epistasis in response to selection is growing, and epistasis may be responsible for an increased, or non-linear rate, of adaptation in natural populations or artificial selected lines[94]. Additionally, having loci under NFDS not physically linked to the Y can shield them from increased drift experienced on the Y, allowing for higher levels of variation to be maintained[95]. Finally, for NFDS to operate, we would hypothesise that there should be a balance between keeping coadapted alleles together and breaking them apart to create variation. Together, this suggests that Y-autosome epistasis acting on a diverse autosomal haplotype presents a feasible hypothesis for the maintenance of colour pattern traits. The Iso-Y lines offer a unique resource to explore interactions between LG1 and LG12, and also to further distinguish the role of the three different regions on LG1 by performing focussed crosses between Iso-Y lines. Future studies should also aim to fully characterise the diversity of LG1, including additional long-read sequencing of multiple individuals from both LP and HP environments in order to determine the reservoir of variable haplotypes present in this region.

## Methods

**Generation of the Iso-Y lines**. All procedures involving live animals were reviewed and approved by the Florida State University Animal Care and Use Committee (protocol no. 1442 and no. 1740). The Iso-Y lines were kindly provided by AE Houde, who established them by choosing male lineages in which colour pattern on the body was strongly Y-linked. Each line was founded by a single male drawn from the 'Houde' tributary of the Paria River in Trinidad (Trinidad National Grid System: PS 896886). Males from this tributary are known to show strong Y-linkage[19]. Each Iso-Y line was maintained by breeding males with colour patterns similar to that of each line's founder; hence, the colour patterns of the Iso-Y lines are ecologically relevant. Every generation, males were mated to females sired by males from the same line, or, every 2–3 generations, backcrossed into the stock population derived from the same Houde tributary. The lines have been maintained at Florida State University since 2012.

### Colour pattern phenotype analysis

*Photography.* We used multispectral digital photography (Sony A7 with full-spectrum conversion; Nikon 80 mm f/5.6 El-Nikkor Enlarging Lens) to capture human-visible and ultraviolet images of males from each Iso-Y line (Iso-Y6$_n$ = 41; Iso-Y8$_n$ = 48; Iso-Y9$_n$ = 42; Iso-Y10$_n$ = 42). Fish were lightly anesthetised by immersion in a Tricaine mesylate (Pentair) solution and placed on a clear petri dish above a grey background with the left side of the body facing upwards. A soft tip miniature paint brush was used to raise the dorsal fin, flare the caudal fin, and lower the gonopodium so that these appendages were visible. Fish were illuminated by four metal halide lights (Hamilton, 6500 K bulbs) that simulate the natural photic environment. A size standard and two full-spectrum colour standards (grey—20% reflectance, white—99% reflectance; Labsphere) were included beside the fish. Glare and shadow were minimised by placing a diffuser (cylinder of 0.015" polytetrafluoroethylene) around the fish and colour standards. We photographed each fish once in the human visible spectrum (Baader UV/IR cut / L-Filter) and once in the UV spectrum (Baader U-Venus-Filter 350 nm).

*Morphometrics.* Morphometrics were performed using the TPS Series software[96], using tpsUtil v1.81. We used tpsDig2 v2.31[97] to set the image scale using the size standard, and to place landmarks around the perimeter of the fish. Following Valvo et al.[39], we placed seven traditional landmarks at the tip of the snout, the anterior dorsal fin attachment, the posterior dorsal fin attachment, the dorsal caudal fin attachment, the ventral caudal fin attachment, the posterior gonopodium attachment, and the anterior gonopodium attachment. We then placed 55 semi-landmarks at approximately even intervals between the traditional landmarks. We used sliding of semi-landmarks to minimise any shape variation resulting from unequal distribution of semi-landmark placement. Next, tpsSuper v2.05[96] was used to generate a consensus shape representing the average shape of the males in all 346 photos (173 males * 2). Images of each individual were unwarped to this consensus shape, thereby mapping every pixel from the original image to an analogous location on the consensus shape.

*Colour measurement.* Analysis using Colormesh v2.0[39] was performed in R v4.0.2. We first performed Delaunay triangulation, which is used to reconstruct a complex shape (i.e. the shape of the unwarped fish) using a concise number of points distributed across the surface of that shape. We then measured the average colour of pixels in a radius around each of these sample points. We used cross-validation to determine the optimal number of Delaunay triangulations (more triangulations result in more granular colour pattern data) and sample circle radius (see below). At each sample point, we extracted linear colour measurements for four colour channels: R (red), G (green), and B (blue) and UV. We accounted for any minor fluctuations in the lighting environment by calibrating the colour values for each channel by subtracting the average deviation of colour measured in the photo on the white and grey colour standards from the known reflectance values of those standards.

*Colour pattern differences among Iso-Y lines.* We used Discriminant Analysis of Principal Components (DAPC) to describe the properties of colour pattern as this analysis is recommended for characterising the properties of groups using high dimensional data sets. We used the dapc function in adegenet v2.1.3[40] to perform a Principal Components Analysis (PCA), followed by a discriminant analysis to define the linear combinations of PC scores that minimise within and maximise between group variances.

We used DAPC in a cross-validation framework to determine the scheme for capturing colour data that allowed us to best discriminate among the Iso-Y lines[39]. We used cross-validation to determine the optimal number of PC's to retain, number of Delaunay triangulations to perform, and sample circle radius. Using the xvalDapc function, we performed DAPC on training and validation data, and examined the average proportion of successful assignments for a varying number of retained PC's. We defined the training and validation populations each as 50% of the individuals from each line. We set the maximum number of PC's to retain for cross-validation as *n.pca* = 173 (the number of fish), with cross-validations performed at 17 different retention levels in increments of 10 PC's (up to 170). We performed 100 replicates per PC retention level. The average proportion of successful placements was maximised (at 98.8% successful) by retaining 10 PC's, using four Delaunay triangulations, and a sample circle radius of one pixel. Consequently, we used this sampling scheme to measure colour pattern for all phenotypic analyses, retaining 10 PC's for DAPC. This scheme resulted in 9904

colour measurements per fish (four colour channels * 2476 sampling locations), with colour averaged over 5 pixels per sample point.

To visualise the colour variation summarised by the discriminant functions, we generated heat maps depicting the correlations between position on each discriminant function and colour for each colour channel at each sampling location.

*Comparing colour pattern differences among Iso-Y lines.* To determine whether the Iso-Y lines have robust phenotypic differences, we used a permutational MANOVA to compare mean colour measures among lines. We first reduced the dimensionality of the colour pattern data for this analysis using PCA. We retained the first 17 PC's to summarise the greatest amount of variation in the data while keeping the ratio of observations to variables greater than 10:1 for our subsequent multivariate analyses. These PC's together explained 59.3% of the total variation in colour measurements. We then compared PC scores among lines using the adonis function in vegan v2.5-6[98], which computes the pairwise distances among observations and then performs a permutation test (randomly assigning labels among factors) to partition the distance matrix among sources of variation. P-values were then calculated using two-sided pseudo-F ratio tests. We used Euclidean distance and created 10 000 permuted samples per test. We additionally visualised the differences in phenotype among Iso-Y lines by generating density plots of each Iso-Y line for all 17 PC's used in permutational MANOVA (Supplementary Fig. 2).

*Comparing colour pattern variance among Iso-Y lines.* To quantify total within-line variance in colour pattern, we calculated the trace of the variance-covariance matrix among fish within a given line (across all colour channels and sampling locations). We then used permutational t-tests to determine whether phenotypic variance differed between each pair of lines, using the sample function. We permuted the line labels for each whole-fish colour pattern, thereby accounting for the fact that all colour measures on the same fish are not independent of one another. We created 10 000 permuted samples per test and performed two-tailed tests.

**Genomic library preparation and variant genotyping**

*Pool-seq of the Iso-Y lines.* Genomic DNA was extracted from 48 males of each Iso-Y line using an ammonium acetate extraction method from caudal peduncle[99]. DNA quality was assessed using gel electrophoresis and DNA concentration calculated using Quant-iT Picogreen dsDNA reagent (Invitrogen) on a Glomax Explorer Microplate reader (Promega). DNA of each individual was diluted to 18-22 ng/µl and 4 µl of each added to a pooled sample per Iso-Y line. Pools were cleaned using a NEB Monarch clean up kit, with final concentration and quality of each confirmed using a Nanodrop ND-1000 (Thermo Fisher) and a Bioanalyser (Agilent). Library preparation and sequencing was performed at the University of Exeter Sequencing Service, using a NextFlex RAPID PCR-free library preparation protocol. Each of the libraries were sequenced across multiple lanes on a HiSeq 2500 in standard mode, with a 125 bp paired-end metric (Supplementary Table 9).

Raw reads were cleaned using cutadapt v1.13[100]. Reads were aligned to the guppy genome[31] (https://www.ebi.ac.uk/ena/browser/view/GCA_904066995) with bwa mem v0.7.17[101] and converted to sorted bam alignment format with samtools v1.9[102]. Coverage was calculated using qualimap v2.2.1[103]. Variants were called using Freebayes v1.3.1[104] with GNU parallel[105] by chunking the alignment files into regions based on coverage using sambamba v0.7[106]. Freebayes was run with the options: --use-best-n-alleles 4 -g 1000. The raw variant output was filtered for biallelic SNP variants at QUAL > 30 and DP > 10, followed by a maximum missing filter of 80% applied to each pool separately followed by the application of a minor allele frequency (maf) of 25% applied across all pools using vcftools v1.9[107]. The VCF file was used as input to poolfstat v1.2[108], where the variants were filtered to exclude sites at <30× minimum coverage and <500× maximum coverage per pool, with a minimum read count per allele of 10 (Final SNP set = 3,995,905 SNPs). We used the curated liftover for LG12 for all analyses and plots of LG12[109].

*Whole-genome sequencing (WGS) of wild-caught individuals.* For the WGS of 26 wild-caught guppies, we sequenced individuals from the Paria river (n = 9) and used these with previously available data[31] (Supplementary Data 5). Genomic DNA was extracted using the Qiagen DNeasy Blood and Tissue kit (QIAGEN, Cat No./ID: 69506). DNA concentrations ≥35 ng/µl were normalised to 500 ng in 50 µl and were prepared as Low Input Transposase Enabled (LITE) DNA libraries at The Earlham Institute, Norwich. LITE libraries were sequenced on an Illumina HiSeq4000 with a 150 bp paired-end metric and a target insert size of 300 bp, and were pooled across several lanes so as to avoid technical bias with a sequencing coverage target of ≥10× per sample. Data from the LP Marianne river was previously generated (n = 17)[31]. Although the Paria and LP Marianne guppies are sampled from different sites, there is strong evidence that gene flow occurs between the populations occupying the upper reaches of these rivers[110,111]. We found that genome-wide mean $F_{ST}$ calculated between populations in this dataset was 0. Importantly, there was no effect of river on haplotype structure (Supplementary Data 5). Data processing followed previous methods using GATK4 v4.1.8.1[112]. Final filtering of the VCF file included filtering for bi-allelic SNPs at a minimum depth of 5 and a maximum depth of 200, removing 50% missing data and application of a 10% maf filter (Final SNP set = 1,021,495). Variants were phased

individually with Beagle v5.0[113], and then phased again using Shapeit v2.r904[114,115] making use of phase-informative reads (PIR)[116]. We used the curated liftover for LG12 for all analyses and plots of LG12[109].

*Long-read sequencing.* To assist with the phasing of variants called from short-read data and to detect structural variants (SVs) we generated 20 Gb of long-read Pacbio data from one of the Iso-Y individuals (Iso-Y6). High molecular weight DNA was extracted using DNeasy Blood & Tissue Kit (QIAGEN, Cat No./ID: 69506) with modifications (10× Genomics Sample Preparation Demonstrated Protocol and MagAttract HMW DNA Kit handbook). Data were sequenced on 3 SMRT cells of a PacBio Sequel at the University of Exeter Sequencing Service. For phasing, reads were aligned to the guppy genome using minimap2 v2.17[117]. Genotypes were phased using whatshap v0.18[118] using the reference genome and the long-reads to lift phasing information.

**Analysis of Iso-Y-line genomic data.** Analyses were conducted in R v4.0.1. Pairwise $F_{ST}$ was calculated using the Anova $F_{ST}$ method implemented in poolfstat v1.2[108]. Per SNP $F_{ST}$ values were compared pairwise between each Iso-Y line. To identify chromosomes with the highest mean variance in $F_{ST}$ differentiation across our dataset, mean Z-scores of each PC were summarised for each chromosome[42]. These were calculated using the prcomp function using stats v3.6.2, centering and scaling the results.

To delineate line-specific blocks apparent from the distinct patterns of differentiation observed in the $F_{ST}$ analysis, we evaluated the allele frequencies (AFs) of each line. AFs were extracted from the Pool-seq object generated by poolfstat and were polarised to Iso-Y9 (the line which showed the highest genetic differentiation). To explore delineation breakpoints in the AFs, we adopted the use of change point detection (CPD) analysis. CPD was conducted in R using changepoint v2.2.2[119]. We used individual AFs from each Iso-Y line as input to detect mean changes, using the BinSeg method and SIC criterion. The number of changepoints identified is Q; in cases where several change points were detected, we increased the value of Q to 10. To add support to the change points detected using the AFs, and to ensure we were not missing any additional breakpoints, we also used PC1 $F_{ST}$ scores as input. In cases where multiple change points were detected within close vicinity of one another, caution was taken to delimit the smallest region in each case; this was to correctly identify the minimum unit of inheritance. Within the identified CPD regions, we further inspected the segregation of Iso-Y line alleles responsible for driving differentiation by plotting the AFs of each identified region[120].

To assess diversity among the Iso-Y lines, π was calculated from the Iso-Y line allele frequencies on a per base pair basis[121]. To summarise the among-Iso-Y line variation, per SNP π values were used as input to PCA, calculated using the prcomp function.

For functional gene annotation, we extracted the regions of interest from the guppy genome (https://www.ebi.ac.uk/ena/browser/view/GCA_904066995) using samtools faidx (v1.9)[102] and aligned them to the previous guppy genome assembly (https://www.ebi.ac.uk/ena/browser/view/GCA_000633615.2) using minimap2 v2.17[117] and pulled the uniprot gene IDs from the annotation using biomaRt v2.44[122]. GO enrichment and KEGG mapping were performed using clusterProfiler v3.18.1[123]. For KEGG mapping, we used Ensembl's gene dataset for the species as the universe. For GO enrichment, we used the AnnotationHub v2.22.1 to pull the latest available OrgDb (AH86018).

**Analysis of the natural population WGS data.** WGS data comprising 26 wild-caught individuals was used to explore the identified Iso-Y regions in natural guppy populations. Linkage disequilibrium (LD) was calculated among polymorphic SNPs for LG1 and LG12. Invariant positions were removed using bcftools v1.8[124], variants were thinned at 5 kb intervals and r2 values were calculated in plink v1.9; bwh.harvard.edu/plink, outputting a square matrix for plotting using LDheatmap v1.04[126].

Shifts in localised heterogeneity have been explained as a potential artefact of chromosomal inversions or long maintained haplotypes. We analysed patterns arising from changes in local ancestry in the natural data using lostruct v0.9[65]. Local PCAs were run with three PCs mapped onto three MDS. Mapped eigenvalues > 0.01 were assessed for saturation before defining significant MDS. Significant outlier windows of each MDS were defined by first calculating 3 standard deviations of the mean of the MDS distribution after trimming the 5% tails of each distribution, followed by returning all windows at the extremes of the distribution. To balance loss of power and sensitivity, local PCAs were assessed in both 10 bp and 100 bp windows (Supplementary Fig. 24). We considered the start and end positions of the signature of each MDS as the last region where we found 3 or more adjacent neighbouring windows.

Population genetics statistics were calculated along LG1 and LG12 using PopGenome v2.7.5[127]. Intersex-$F_{ST}$, $D_{xy}$ and π were calculated in non-overlapping 1 kb windows. Intersex $D_a$ was calculated as $D_{xy}$ - female π. Outliers were considered if the regions were outside of the upper and lower 95% quantiles of each calculated statistic. GC and repeat content were calculated from the male guppy genome[31], computed in 1 kb windows.

**Analysis of structural variants and coverage**. For the medium-coverage WGS data derived from wild-caught individuals we used BreakDancer v1.4.5[128] and Lumpy-based smoove v0.2.5[129] using SVtyper v0.7.0[130]. We also applied these two short-read methods to the Iso-Y Pool-seq data, in addition to Manta v1.6.0[131] as the data were high-coverage and PCR-free. SV outputs were filtered depending on the software: BreakDancer SVs were filtered at a confidence score > =99, read group (RG) support > =median RG; manta SVs were filtered at a quality score > = 999 and only events where both paired-read and split-read support at values realistic of the overall read depth were retained; for Lumpy/smoove SVs, events that had evidence of both split read and paired end support at realistic read depth values were considered. We excluded SVs < = 10 kb. We also made use of our long-read sequencing data from one Iso-Y line individual (Iso-Y6) using sniffles v1.0.12[132] and PBSV v2.3.0 (https://github.com/PacificBiosciences/pbsv). Any suspected SVs were manually inspected in the Integrated Genomics Viewer (IGV v2.4.8;[133]).

Coverage was estimated from bam files using the *bamCoverage* option of deeptools v3.3.1[134]. The first and last 100 kb of each chromosome were trimmed before estimating coverage with a binSize = 50, smoothLength = 75, and an effectiveGenomeSize of 528,351,853 bp (genome size minus masked and trimmed regions). The ends of chromosomes were trimmed because they often showed high peaks of coverage, thereby distorting the normalised measures across the chromosome. Coverage estimates were normalised using RPGC (reads per genomic context). Bins with normalised coverage >4 (four times the expected median of 1) were filtered. For the natural WGS data, coverage was averaged within populations. Outputs were converted to bed files with window sizes of 1 and 10 kb by taking weighted means.

**Analyses of multiple haplotypes on LG1**. Genotype and haplotype plots were created using Genotype Plot in R[135]. For the haplotype-based analysis, we polarised the phased haplotypes to Iso-Y9. Breakpoints between the phases were quantified by identifying the location (in bp) where phase0 switched to phase1, and vice versa. Switchpoints were quantified in males and females and a switchpoint was considered to be conserved when the locations (within 50 kb) overlapped in two or more individuals.

Inter-chromosomal linkage was calculated as above for intra-chromosomal linkage calculating r2 values between LG1 and LG12 using the --inter-chr flag in plink v1.9, outputting a normal pairwise matrix. Results were plotted in R using custom scripts[120].

**Reporting summary**. Further information on research design is available in the Nature Research Reporting Summary linked to this article.

## Data availability
The DNA sequencing data generated in this study have been deposited in the European Nucleotide Archive (ENA) under the Study Accession PRJEB36506 with the following codes: SAMEA6512722-SAMEA6512725 (Pool-seq Iso-Y data); SAMEA8750557-SAMEA8750565 (whole-genome sequencing data for Paria); SAMEA8795870-SAMEA8795872 (long-read pacbio data for Iso-Y6). The DNA sequencing data generated for Upper Marianne individuals have been deposited in the ENA under the Study Accession PRJEB10680 under the accession codes: SAMEA3649957-SAMEA3649973. The male guppy reference genome can be accessed at the ENA under the Accession GCA_904066995. The female guppy reference genome can be accessed at the ENA under Accession GCA_000633615.2. Source data are provided with this paper at https://github.com/josieparis/guppy-colour-polymorphism; https://doi.org/10.5281/zenodo.5036659. Recombination data were provided through personal communication with permission of the authors.

## Code availability
All code, scripts and additional data related to analysis are available on GitHub (https://github.com/josieparis/guppy-colour-polymorphism; https://doi.org/10.5281/zenodo.5940510), (https://github.com/josieparis/gatk-snp-calling; https://doi.org/10.5281/zenodo.5903522) and (https://github.com/JimWhiting91/genotype_plot; https://doi.org/10.5281/zenodo.5913504).

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

## Acknowledgements

We wish to thank Anne Houde for the initial collection of the Iso-Y line fish from the Paria River, and Helen Rodd for the collection of the wild-caught fish from the Paria River. We also wish to thank Jennifer Valvo for assistance in the maintenance of the Iso-Y lines, and Sally Lepzinski for helping to photograph fish. Thanks to Deborah Charlesworth for experimental insight and suggestions. Computational infrastructure support was provided by The University of Exeter's High-Performance Computing (HPC) facility (ISCA). DNA sequencing was performed by the University of Exeter Sequencing Service (ESS). The project was funded by the Natural Environment Research Council (NERC, NE/P013074/1) (J.R.P., B.A.F.), EU Research Council grant (GuppyCon 758382) (B.A.F., J.R.W., M.v.d.V.) and the National Science Foundation of the United States (NSF) ISO-1354775 and DEB-1740466 (M.J.D., K.A.H.).

## Author contributions

J.R.P. carried out molecular work for the WGS data, performed genomic and statistical analysis, interpretation, and wrote the manuscript. J.R.W. assisted with analysis and interpretation throughout. M.J.D. conducted the breeding design, performed phenotyping analysis and wrote parts of the manuscript. J.F.O. assisted with analysis, interpretation and figure preparation. P.J.P. performed the molecular lab work for the Pool-seq data. M.v.d.Z. assisted with molecular work and analysis. C.W.W. assisted with analysis of the Pool-seq data. K.A.H. conceived the project, oversaw the breeding experiments, provided analysis and interpretation throughout. B.A.F. conceived and supervised the project, helped with analysis, and co-wrote the manuscript. All authors provided comments on earlier drafts.

## Competing interests

The authors declare no competing interests.
