## [Peer Review File · Nature Communications]

A large and diverse autosomal haplotype is associated with sex-linked colour polymorphism in the guppyREVIEWER COMMENTS

Reviewer #1 (Remarks to the Author):

This is a very well-done and well-written study of the genetic basis of male color patterns in guppies, which have shown to be sexually antagonistic and under negative frequency dependent selection. Although these color patterns have classically been thought to be Y-linked, this study conclusively demonstrates that divergent haplotypes on an autosome controls these color patterns, at least in this population of guppies. This is an important contribution to the field of sex chromosome evolution and adds clarity to recent controversies about the role of sexually antagonistic selection in the evolution of guppy sex chromosomes. This study further lays the groundwork for many interesting follow-up studies on these iconic color patterns.

I only have one major comment and some minor editorial comments for the authors:

Although the evidence is convincing for epistasis between the Y chromosome and chromosome 1, I still think the authors need to consider the possibility that there has been a translocation event in the lab populations. Because they mapped sequences to the reference genome, such a translocation would not be identified (and computational methods to do so are not that reliable). And, to my knowledge, genetic linkage maps have not been made in these populations. The LD data in the natural populations argues against such a translocation, but evolution of sex determination mechanisms and sex chromosomes have occurred in lab-adapted populations in a similar number of generations in both zebrafish and *Xiphophorus* (Wilson et al 2014 *Genetics* 198: 1291-1308; Franchini et al 2018 *Nature Communications* 9: 5136). This caveat should be mentioned in the discussion (probably at L395-396). Furthermore, another more explicit explanation for the Y-autosome epistasis is simply that the color patterns are only expressed in males (due to testosterone). This should also be discussed.

Minor comments

L25: reiculata should be reticulata

L101: I did not see p-values in Supplementary Table 1.

L120-122: I was a little confused here because the writing implies that you are discussing the percentage of total SNPs above the critical score of 3 that are found on different chromosomes or scaffolds. But in Supplementary Figure 4b, there are scaffolds with 0.8 – are these scaffolds not linked to chromosomes? Do you mean 0.8% or a frequency of 0.8? This should be clarified in the figure legend.

L145: Can you add shading to Supplementary Figure 6 to indicate the three regions of differentiation?

L167: If there is space, I would just add the short Supplementary Text 1 here. It's annoying to need to go to another document for such a brief description.

L183: The colors indicated in the figure legends for Supplementary Figures 13 and 14 are incorrect.

L189: I think Supplementary Figure 15 is very useful and could be added to the main text.

L191-199: Again, the Supplementary Text 2 is not much longer than this paragraph and it could be substituted here, as it nicely discusses candidate genes in all three regions.

L272: Is it possible that these NLRP1-related genes are related to sex determination?

L274-275: By Y candidates do you mean candidate sex determination genes? I would be more explicit with the wording.

Figure 4: It would be interesting to have color phenotype data for these recombinant males!

Signed by Catherine Peichel

Reviewer #2 (Remarks to the Author):

The study by Paris et al aims to identify the genetic architecture of sex-linked colour polymorphism in the guppy. The guppy is a great model for sexual selection and sex chromosome research, and has

been the subject of considerable recent interest. Therefore, the study is addressing a really interesting question. However, I have a number of concerns about the manuscript both in terms of the presentation but also the results. My primary concern is a lack of control for the iso-Y lines. Without this it is difficult to infer that the regions identified are associated with colour or just reflect segregating variation in the founder lines that has been preserved due to low local recombination rates. My specific comments are below:

I felt that the characterisation of the state of the field and novelty of the results was imprecise. The framing in the abstract, introduction and discussion leads the reader to understand that colour genes are assumed to be sex-linked and part of a supergene on the sex chromosomes. However, a study recently showed that guppy colouration is not predominantly Y-linked and suggest epistatic effects and Y-linked modifiers are important (Morris et al 202 Proc Biol Sci). The findings of this paper are extremely relevant to this study, however, it was only referenced in passing once. Therefore, while the authors provide an advance in this area by identifying the genetic architecture of colour traits, the idea that epistatic effects between the autosomes and Y is not novel. This should be fully acknowledged and the novelty of these results conveyed more accurately.

I also found the description of guppy sex chromosome divergence to be inaccurate (L44 onwards). While I appreciate there is debate over the extent of recombination suppression between the X and Y, the representation of the state of the field was rather one sided. There are a number of studies from the Mank lab showing a wider region of recombination suppression and Y-linkage that are excluded from the references (see below). Given the weight of evidence, this must be acknowledged fully. Additionally, the authors statement that there are 'many segregating male-specific variants' seem to contradict their statement in the intro that the 'Y-region must be small, possibly only a single gene'.

Almeida P, Sandkam BA, Morris J, Darolti I, Breden F, Mank JE (2021) Divergence and remarkable diversity of the Y chromosome in guppies. *Molecular Biology & Evolution* 38: 619-633

Darolti I, Wright AE, Sandkam BA, Morris J, Bloch NI, Farré M, Fuller RC, Bourne GR Larkin DM, Breden F, Mank JE (2019) Extreme heterogeneity in sex chromosome differentiation and dosage compensation in livebearers. *Proceedings of the National Academy of Sciences, USA* 116: 19031-19036

Darolti I, Wright AE, Mank JE (2020) Guppy Y chromosome integrity maintained by incomplete recombination suppression. *Genome Biology & Evolution* 12: 965-977

Morris J, Darolti I, Bloch NI, Wright AE, Mank JE (2018). Shared and species-specific patterns of nascent Y chromosome evolution in two guppy species. *Genes* 9: 238

The authors identify regions exhibiting sequencing divergence between the Iso-Y lines and conduct a number of tests. They then conclude that a region on LG1 is important for sex-linked colour polymorphism. I think this is an interesting idea, however, in my opinion, the weight of evidence is not sufficiently strong and there are a number of additional tests that should be implemented:

1. The authors find evidence of regions on LG1 that differ between iso-Y lines. Apologies if I misunderstood but this in itself isn't evidence that these regions are important for sex-linked colour polymorphism. If a region of low recombination is naturally segregating in the population (and the results suggest that are multiple haplotypes in the source pop) by chance each individual male chosen to start each iso-Y line could have a different haplotype. If the region doesn't recombine frequently, then you would expect to see the haplotype preserved in each line. The presence of additional haplotypes could reflect haplotypes from the source population that have been introgressed recently by backcrossing. Ideally, there would have been replicates of each iso-Y line or at least a control line to disentangle this effect. Clearly, I appreciate this isn't possible now, but in my opinion, it means the results should be treated with caution.

2. Previous research has shown that recombination between the X and Y chromosome occurs very infrequently. Therefore, similar to the point above, it is reasonable to assume that Y chromosomes in each iso-Y line are not recombining with the X from the stock population as often as autosomes are. I think that the infrequent recombination between the Y in the iso-Y line and the X from the stock

population, coupled with genetic drift between the iso-Y lines, would result in greater sequence divergence of LG12 between lines. Furthermore, we would expect this to affect the length of LG12. This is exactly what the study finds. However, the authors instead suggest this indicates multiple Y-haplotypes associated with colour polymorphism. How can the authors distinguish between these scenarios and the more neutral explanation I explain above?

3. The authors identify candidates for colour and male-specific fitness on LG1 (L191). This finding would be strong support for LG1 playing a role in colour polymorphism. However, it is not clear whether these regions are actually significantly enriched for these types of loci? Without this, the weight of evidence is not strong and the results are rather speculative.

4. The authors calculate intersex F_{st} across LG1 and identify a region of high differentiation (L230). They suggest that these results indicate differential selection pressures. However, the use of intersexual F_{st} to infer differential selection has been heavily criticised. First, it has been shown that much of the signal in previous work is driven by recent Y-linked duplicates (e.g Bisseger et al 2020 *Mol Ecol*), however, this appears not to be the case here as male-specific diversity is not elevated. Second, the selection coefficients required to drive elevated intersexual F_{st} observed in previous work has been shown to be implausible (e.g Kasimatis et al 2019 *G3*). At a minimum, the authors should estimate the selection coefficients and selection load necessary to generate the intersexual F_{st} values they observe to test whether they are indeed plausible.

Bisseger, M, Laurentino, TG, Roesti, M, Berner, D. Widespread intersex differentiation across the stickleback genome – The signature of sexually antagonistic selection? *Mol Ecol*. 2020; 29: 262– 271. <https://doi.org/10.1111/mec.15255>

Katja R Kasimatis, Peter L Ralph, Patrick C Phillips, Limits to Genomic Divergence Under Sexually Antagonistic Selection, *G3 Genes|Genomes|Genetics*, Volume 9, Issue 11, 1 November 2019, Pages 3813–3824, <https://doi.org/10.1534/g3.119.400711>

5. The authors use intersexual F_{st} on LG12 in the natural source population as support for multiple diverse Y haplotypes (L269, Fig 3). I don't follow this - elevated F_{st} and an excess of male diversity are common signatures of recombination suppression between sex chromosomes. They are used to identify sex-linked regions not multiple Y haplotypes?

Minor comments

L44 'Through multiple independent population genomic and pedigree crossing studies, it can be concluded that the Y-specific region must be small, possibly only a single gene, occurring near the distal end of chromosome twelve (LG12) 30–32 (but see 33,34).' I do not think this is a fair representation of the literature - see point at the beginning of the review

L46 I also 'No colour or sex candidate genes have been identified in this region, and moreover, the entire LG12 is not enriched with colour genes' I also do not think this is a fair representation. Morris et al 2018 found a significant enrichment of possible pigmentation genes in their Y gene assembly.

L49 'However, intriguingly, there is evidence that the candidate Y region is highly diverse among males with many segregating male-specific variants, indicative of multiple Y haplotypes' This seems inconsistent with the earlier statement that the Y-specific region might only contain a single gene.

L55 'These data suggest that colour-pattern genes are not physically linked to the SDL, but may be regulated by sex-specific loci.' This has been shown before by Morris et al 2020 and so should be acknowledged.

L245 onwards. As mentioned above, previous research that should be cited is not. For instance, the patterns of intersexual F_{st} reported are almost identical to Almeida et al 2021 *MBE*.

Almeida P, Sandkam BA, Morris J, Darolti I, Breden F, Mank JE (2021) Divergence and remarkable diversity of the Y chromosome in guppies. *Molecular Biology & Evolution* 38: 619-633

L421 This statement is incorrect. Wright et al 2017 results were corroborated by independent data in Almeida et al 2021.

Reviewer #3 (Remarks to the Author):

The manuscript "A large and diverse autosomal haplotype is associated with sex-linked colour polymorphism in the guppy" presented by Paris et al. uses a combination of genomics techniques and selective breeding to examine what other regions of the genome may be interacting with the Y chromosome to determine male coloration. This is an intriguing study with important implications because guppy coloration helped form some of the fundamental hypotheses of Y linked inheritance in the early 20th century, yet many studies have shown that much of male guppy coloration is not inherited in a perfect Y manner. This makes this a compelling system to begin to test how selection can act on phenotypes that are highly polygenic. I appreciate the tremendous amount of work and wide variety of genomic and population genetic analyses the authors use to ask where in the genome variation lies that corresponds to colour variation.

From the data presented here I'm convinced there is variation in LG1 that influences male coloration. However, I'm still left wondering about the authors interpretation about why this suggests epistasis between LG1 and LG12? LG12 is known to be the sex determiner, and the breeding design was to create iso-Y lines, but females don't ever express color so the phenotype was never assessed outside the context of the iso-Y line. It seems to me it's still quite possible that LG1 is just behaving as a normal autosomal locus, and epistasis with LG12 isn't necessary to explain this? The signals of decreased diversity on LG12 are absolutely expected from the breeding design since this was the creation of an iso-Y line, and while the LG1 results indicate there is likely something underlying coloration, I'm not convinced this is an epistatic interaction. It would be interesting to see how much of colour inheritance is on LG1 by treating females from these lines with testosterone and comparing their expressed color pattern.

L. 387 (and several points throughout that form the framework on this study). Authors bring up the idea that sexual conflict is expected to move colour genes to the Y chromosome in guppies. However, the theory of sexual conflict does not actually apply to modern guppy coloration (a longstanding misunderstanding in guppy literature). Sexual conflict exists when an allele is beneficial to one sex but detrimental to the other sex. But there are two ways to resolve sexual conflict: either by moving genes to the sex-limited chromosome (as proposed here) or by sex limited gene regulation- that is to say by expressing that gene only in the sex which benefits from it. Once conflict is resolved (such as the evolution of gene regulation) then sexual conflict is no longer exerting selection to move genes to the sex-limited chromosome. In guppies – expression of coloration genes depends entirely on testosterone (as evidence by numerous studies, included citations in this the paper, that show females express colour genes when given testosterone) and thus color genes in guppies are not under sexual conflict.

Several points throughout the manuscript (including abstract, introduction and discussion) the authors suggest that the field believes most of the color genes in guppies are on the Y (eg. L. 347). However, this is a misrepresentation as even the original work by Fisher showed only components of guppy color patterns are Y inherited. To my knowledge there has never been a study of copy coloration that suggests that ALL male coloration is entirely inherited on the Y.

REVIEWER COMMENTS

Reviewer #1 (Remarks to the Author):

This is a very well-done and well-written study of the genetic basis of male color patterns in guppies, which have shown to be sexually antagonistic and under negative frequency dependent selection. Although these color patterns have classically been thought to be Y-linked, this study conclusively demonstrates that divergent haplotypes on an autosome controls these color patterns, at least in this population of guppies. This is an important contribution to the field of sex chromosome evolution and adds clarity to recent controversies about the role of sexually antagonistic selection in the evolution of guppy sex chromosomes. This study further lays the groundwork for many interesting follow-up studies on these iconic color patterns.

I only have one major comment and some minor editorial comments for the authors:

Although the evidence is convincing for epistasis between the Y chromosome and chromosome 1, I still think the authors need to consider the possibility that there has been a translocation event in the lab populations. Because they mapped sequences to the reference genome, such a translocation would not be identified (and computational methods to do so are not that reliable). And, to my knowledge, genetic linkage maps have not been made in these populations. The LD data in the natural populations argues against such a translocation, but evolution of sex determination mechanisms and sex chromosomes have occurred in lab-adapted populations in a similar number of generations in both zebrafish and Xiphophorus (Wilson et al 2014 Genetics 198: 1291-1308; Franchini et al 2018 Nature Communications 9: 5136). This caveat should be mentioned in the discussion (probably at L395-396). Furthermore, another more explicit explanation for the Y-autosome epistasis is simply that the color patterns are only expressed in males (due to testosterone). This should also be discussed.

Thank you for raising these two important points. We have added that no elevated inter-chromosomal LD was observed in the natural data (lines 538-539) and also text to the Discussion to address this alternative that indeed, a translocation may have occurred in the laboratory (lines 541-543). We have also added text to the discussion regarding the expression of colour genes to (lines 493-503), as also pointed out by Reviewer 3.

Minor comments

L25: reiculata should be reticulata

Corrected (line 60).

L101: I did not see p-values in Supplementary Table 1.

These have now been added.

L120-122: I was a little confused here because the writing implies that you are discussing the percentage of total SNPs above the critical score of 3 that are found on different chromosomes or scaffolds. But in Supplementary Figure 4b, there are scaffolds with 0.8 – are these scaffolds not linked to chromosomes? Do you mean 0.8% or a frequency of 0.8? This should be clarified in the figure legend.

Yes, the scaffolds are unplaced in the genome and we have added this information to the Figure legend. We are discussing percentage of SNPs>critical calculated as $(ncrit_per_chrom/total_ncrit)*100$. We have adjusted the y-axis to read “%” instead of “percentage” for clarity.

L145: Can you add shading to Supplementary Figure 6 to indicate the three regions of differentiation?

Done.

L167: If there is space, I would just add the short Supplementary Text 1 here. It's annoying to need to go to another document for such a brief description.

We have now added this into the main text on lines 227-241.

L183: The colors indicated in the figure legends for Supplementary Figures 13 and 14 are incorrect.

Thank you for spotting this. The colours have now been corrected.

L189: I think Supplementary Figure 15 is very useful and could be added to the main text.

This has now been added as a main Figure (Figure 3).

L191-199: Again, the Supplementary Text 2 is not much longer than this paragraph and it could be substituted here, as it nicely discusses candidate genes in all three regions.

This information has been added into the main text on lines 287-311.

L272: Is it possible that these NLRP1-related genes are related to sex determination?

We previously explored the possibility that the NLRP1-related genes are related to sex determination (Fraser et al 2020 GBE, <https://doi.org/10.1093/gbe/evaa187>). NLRP1 genes form part of the inflammasome and have no (known) link to sex determination. We have added text and a reference to lines 389-392 to expand on the role of this gene for clarity in this particular manuscript.

L274-275: By Y candidates do you mean candidate sex determination genes? I would be more explicit with the wording.

Whilst ‘Y candidates’ might not be explicit; these genes are also unlikely to be sex-determination genes. Instead, we have termed them, “male-trait” candidates and we have expanded slightly on the full names of the genes so the relationship of these genes to “maleness” is more apparent (lines 395-396).

Figure 4: It would be interesting to have color phenotype data for these recombinant males!

Agreed! We have also added this to the discussion on lines 525-528.

Signed by Catherine Peichel

Reviewer #2 (Remarks to the Author):

The study by Paris et al aims to identify the genetic architecture of sex-linked colour polymorphism in the guppy. The guppy is a great model for sexual selection and sex chromosome research, and has been the subject of considerable recent interest. Therefore, the study is addressing a really interesting question. However, I have a number of concerns about the manuscript both in terms of the presentation but also the results. My primary concern is a lack of control for the iso-Y lines. Without this it is difficult to infer that the regions identified are associated with colour or just reflect segregating variation in the founder lines that has been preserved due to low local recombination rates. My specific comments are below:

I felt that the characterisation of the state of the field and novelty of the results was imprecise. The framing in the abstract, introduction and discussion leads the reader to understand that colour genes are assumed to be sex-linked and part of a supergene on the sex chromosomes. However, a study recently showed that guppy colouration is not predominantly Y-linked and suggest epistatic effects and Y-linked modifiers are important (Morris et al 202 Proc Biol Sci). The findings of this paper are extremely relevant to this study, however, it was only referenced in passing once. Therefore, while the authors provide an advance in this area by identifying the genetic architecture of colour traits, the idea that epistatic effects between the autosomes and Y is not novel. This should be fully acknowledged and the novelty of these results conveyed more accurately.

I also found the description of guppy sex chromosome divergence to be inaccurate (L44 onwards). While I appreciate there is debate over the extent of recombination suppression between the X and Y, the representation of the state of the field was rather one sided. There are a number of studies from the Mank lab showing a wider region of recombination suppression and Y-linkage that are excluded from the references (see below). Given the weight of evidence, this must be acknowledged fully. Additionally, the authors statement that there are 'many segregating male-specific variants' seem to contradict their statement in the intro that the 'Y-region must be small, possibly only a single gene'.

Almeida P, Sandkam BA, Morris J, Darolti I, Breden F, Mank JE (2021) Divergence and remarkable diversity of the Y chromosome in guppies. *Molecular Biology & Evolution* 38: 619-633
Darolti I, Wright AE, Sandkam BA, Morris J, Bloch NI, Farré M, Fuller RC, Bourne GR Larkin DM, Breden F, Mank JE (2019) Extreme heterogeneity in sex chromosome differentiation and dosage compensation in livebearers. *Proceedings of the National Academy of Sciences, USA* 116: 19031-19036
Darolti I, Wright AE, Mank JE (2020) Guppy Y chromosome integrity maintained by incomplete recombination suppression. *Genome Biology & Evolution* 12: 965-977
Morris J, Darolti I, Bloch NI, Wright AE, Mank JE (2018). Shared and species-specific patterns of nascent Y chromosome evolution in two guppy species. *Genes* 9: 238

The authors identify regions exhibiting sequencing divergence between the Iso-Y lines and conduct a number of tests. They then conclude that a region on LG1 is important for sex-linked colour polymorphism. I think this is an interesting idea, however, in my opinion, the weight of evidence is not sufficiently strong and there are a number of additional tests that should be implemented:

1. The authors find evidence of regions on LG1 that differ between iso-Y lines. Apologies if I misunderstood but this in itself isn't evidence that these regions are important for sex-linked colour polymorphism. If a region of low recombination is naturally segregating in the population (and the results suggest that are multiple haplotypes in the source pop) by chance each individual male chosen to start each iso-Y line could have a different haplotype. If the region doesn't recombine frequently, then you would expect to see the haplotype preserved in each line. The presence of additional haplotypes could reflect haplotypes from the source population that have been introgressed recently by backcrossing. Ideally, there would have been replicates of each iso-Y line or at least a control line to disentangle this effect. Clearly, I appreciate this isn't possible now, but in my opinion, it means the results should be treated with caution.

We are unclear exactly what the reviewer is suggesting here. We think that the confusion results from the reviewer viewing our lines as an 'artificial selection experiment', where a control line would be needed. The breeding design is better described as an introgression experiment rather than as a 'selection experiment' because females from a randomly-mating population were regularly introduced into all lines. Near fixation of an LG1 region within each line would be very unlikely unless that region contributed to the phenotype that was preserved within each line. Furthermore, with this breeding scheme a 'control line' isn't possible. Any line established from a male and backcrossed into the stock population would also likely differ in Y-linked colour patterns given the strong Y-linkage found in this population. We have updated lines 105-108, line 595-596 to make this clear.

We do, however, take the reviewers point that our results could have been a result of our lab breeding. We discuss the possibility of a lab specific translocation event on lines 541-543 (see also response to reviewer 1). We also therefore chose to compare the Iso-Y line genomes to their natural source. We find no evidence that the Iso-Y lines differ in haplotype structure from the natural population.

2. Previous research has shown that recombination between the X and Y chromosome occurs very infrequently. Therefore, similar to the point above, it is reasonable to assume that Y chromosomes in each iso-Y line are not recombining with the X from the stock population as often as autosomes are. I think that the infrequent recombination between the Y in the iso-Y line and the X from the stock population, coupled with genetic drift between the iso-Y lines, would result in greater sequence divergence of LG12 between lines. Furthermore, we would expect this to affect the length of LG12. This is exactly what the study finds. However, the authors instead suggest this indicates multiple Y-haplotypes associated with colour polymorphism. How can the authors distinguish between these scenarios and the more neutral explanation I explain above?

We agree with the reviewer that some of our results could be driven by variation in recombination across the genome (either on LG1 stated in point 1 or LG12 as stated here). However, we think that the consistent differences shown in the LG1 haplotype and the inconsistent differences found on LG12 are good evidence for LG1 being involved in colour. If LG12 only was involved in colour, then we would predict that there would be at least a small region on LG12 consistently different among the lines. Instead we find that each pairwise difference between the lines occurs at different parts along LG12. This is what you would expect in a low recombining region unrelated to the phenotype and is not the pattern we see within the LG1 haplotype. We have tried to make this point clearer in lines 474-476 and lines 478-485.

3. The authors identify candidates for colour and male-specific fitness on LG1 (L191). This finding would be strong support for LG1 playing a role in colour polymorphism. However, it is not clear whether these regions are actually significantly enriched for these types of loci? Without this, the weight of evidence is not strong and the results are rather speculative.

We have now performed KEGG mapping and GO enrichment on the three regions separately, but our analysis returned few significant or meaningful results after multiple comparison correction. However, when we run all three regions on LG1 together, we get one significant KEGG result for “Lysosome”. This of course, is a top-level term, but as the melanosome is a lysosome-related organelle we feel that this result is potentially interesting. We have added these results to Supplementary Table 10 and to lines 307-311 (also see lines 779-782 for methodology).

We however disagree with the reviewer that enrichment tests are particularly powerful in identifying regions of the genome associated with a specific trait (Maertens et al 2020, Sci. Rep. Article number: 4106). Indeed, these types of analyses are notorious for leading to over-interpreting genomic results and ‘just so stories’ (Pavlidis et al 2012, MBE, 9(10): 3237–3248). Whilst enrichment analyses are sometimes useful to find patterns, we think it is important to note that such analyses miss many important genes, and the matches do not ensure biological relevance. We instead argue that the most important piece of evidence for a genomic region being related to a specific trait is the association between highly differentiated genomic regions and the colour pattern phenotype trait.

4. The authors calculate intersex F_{st} across LG1 and identify a region of high differentiation (L230). They suggest that these results indicate differential selection pressures. However, the use of intersexual F_{st} to infer differential selection has been heavily criticised. First, it has been shown that much of the signal in previous work is driven by recent Y-linked duplicates (e.g Bisseger et al 2020 Mol Ecol), however, this appears not to be the case here as male-specific diversity is not elevated. Second, the selection coefficients required to drive elevated intersexual F_{st} observed in previous work has been shown to be implausible (e.g Kasimatis et al 2019 G3). At a minimum, the authors should estimate the selection coefficients and selection load necessary to generate the intersexual F_{st} values they observe to test whether they are indeed plausible.

Bisseger, M, Laurentino, TG, Roesti, M, Berner, D. Widespread intersex differentiation across the stickleback genome – The signature of sexually antagonistic selection? Mol Ecol. 2020; 29: 262– 271. <https://doi.org/10.1111/mec.15255>

Katja R Kasimatis, Peter L Ralph, Patrick C Phillips, Limits to Genomic Divergence Under Sexually Antagonistic Selection, G3 Genes|Genomes|Genetics, Volume 9, Issue 11, 1 November 2019, Pages 3813–3824, <https://doi.org/10.1534/g3.119.400711>

We agree that inter-sex F_{ST} could be driven by recent duplications or translocation events and not sexual conflict as argued by Bisseger et al. 2020 and Kasimatis et al. 2019. We discuss this possibility in lines 530-531 including the citation of both references outlined above (we added Kasimatis et al 2019). We find no evidence for duplications or translocations on our LG1 haplotype using both short read and long read information.

However, we would like to note that we are not arguing that our candidate region on LG1 is a classic sexual antagonistic locus (see below to our response to reviewer 3). Under sexual antagonism, we would predict that both males and females would have allele frequencies out of HWE at our candidate locus. What we find instead is that it is only males which fall outside of HWE. We hypothesise that a Y-autosome deleterious interaction could be causing this signal (lines 505-507, lines 509-513, lines 525-528). We think including selection coefficients or selection load and taking this analysis further would be too speculative here given our low sample sizes within the population (Ruzicka et al 2020, *Evolution Letters*, 4:398-415). Finally, we have re-written the introduction at lines 49-53,56 and discussion at lines 505-507 & 567-568 to re-frame the context of our study away from sexual antagonism.

5. The authors use intersexual F_{st} on LG12 in the natural source population as support for multiple diverse Y haplotypes (L269, Fig 3). I don't follow this - elevated F_{st} and an excess of male diversity are common signatures of recombination suppression between sex chromosomes. They are used to identify sex-linked regions not multiple Y haplotypes?

Yes, we agree that F_{ST} between the sexes is often used to identify sex-linked regions. However, F_{ST} can be driven by fixed differences between groups or one group being more diverse than the other. Therefore, we coupled F_{ST} with D_a (male specific increased diversity), which can distinguish these signals. We can therefore decipher when F_{ST} is being driven by reduced diversity in males (at our LG1 candidate region) and when it is being driven by increased diversity in males (distal region of LG12). We have updated the writing here on lines 387-389. This is explained more fully in Fraser et al. (2020) GBE in the supplemental methods if the reviewer would like to look at the calculations in more detail.

Minor comments

L44 'Through multiple independent population genomic and pedigree crossing studies, it can be concluded that the Y-specific region must be small, possibly only a single gene, occurring near the distal end of chromosome twelve (LG12) 30–32 (but see 33,34).' I do not think this is a fair representation of the literature - see point at the beginning of the review

We have adjusted the writing here to include relevant references suggested above (see lines 92-94): "Others studies however, conclude that the non-recombining region extends beyond this distal region in some populations based on similar sex-specific genomic diversity measures (Wright et al 2017; Almeida et al 2021).

Wright AE, Darolti I, Bloch NI, Oostra V, Sandkam B, Buechel SD *et al.* Convergent 825 recombination suppression suggests role of sexual selection in guppy sex chromosome 826 formation. *Nat Commun* 2017; **8**: 14251.

Almeida P, Sandkam BA, Morris J, Darolti I, Breden F, Mank JE. Divergence and Remarkable 828 Diversity of the Y Chromosome in Guppies. *Mol Biol Evol* 2021; **38**: 619–633.

L46 I also 'No colour or sex candidate genes have been identified in this region, and moreover, the entire LG12 is not enriched with colour genes' I also do not think this is a fair representation. Morris et al 2018 found a significant enrichment of possible pigmentation genes in their Y gene assembly.

We have added this reference to our introduction at line 80.

L49 'However, intriguingly, there is evidence that the candidate Y region is highly diverse among males with many segregating male-specific variants, indicative of multiple Y haplotypes' This seems inconsistent with the earlier statement that the Y-specific region might only contain a single gene.

Yes, we agree that this is inconsistent. We have updated this paragraph to highlight the different work on delineating the Y chromosome (see lines 88-99).

L55 'These data suggest that colour-pattern genes are not physically linked to the SDL, but may be regulated by sex-specific loci.' This has been shown before by Morris et al 2020 and so should be acknowledged.

We have added this reference to the introduction (line 84).

L245 onwards. As mentioned above, previous research that should be cited is not. For instance, the patterns of intersexual Fst reported are almost identical to Almeida et al 2021 MBE.

Almeida P, Sandkam BA, Morris J, Darolti I, Breden F, Mank JE (2021) Divergence and remarkable diversity of the Y chromosome in guppies. *Molecular Biology & Evolution* 38: 619-633

We have added this reference to line 94, 97, 361 and 553.

L421 This statement is incorrect. Wright et al 2017 results were corroborated by independent data in Almeida et al 2021.

We have adjusted the writing here to reflect this omission (line 553).

Reviewer #3 (Remarks to the Author):

The manuscript "A large and diverse autosomal haplotype is associated with sex-linked colour polymorphism in the guppy" presented by Paris et al. uses a combination of genomics techniques and selective breeding to examine what other regions of the genome may be interacting with the Y chromosome to determine male coloration. This is an intriguing study with important implications because guppy coloration helped form some of the fundamental hypotheses of Y linked inheritance in the early 20th century, yet many studies have shown that much of male guppy coloration is not inherited in a perfect Y manner. This makes this a compelling system to begin to test how selection can act on phenotypes that are highly polygenic. I appreciate the tremendous amount of work and wide variety of genomic and population genetic analyses the authors use to ask where in the genome variation lies that corresponds to colour variation.

From the data presented here I'm convinced there is variation in LG1 that influences male coloration. However, I'm still left wondering about the authors interpretation about why this suggests epistasis between LG1 and LG12? LG12 is known to be the sex determiner, and the breeding design was to create iso-Y lines, but females don't ever express color so the phenotype was never assessed outside the context of the iso-Y line. It seems to me it's still quite possible

that LG1 is just behaving as a normal autosomal locus, and epistasis with LG12 isn't necessary to explain this? The signals of decreased diversity on LG12 are absolutely expected from the breeding design since this was the creation of an iso-Y line, and while the LG1 results indicate there is likely something underlying coloration, I'm not convinced this is an epistatic interaction. It would be interesting to see how much of colour inheritance is on LG1 by treating females from these lines with testosterone and comparing their expressed color pattern.

Thank you for the suggestion; treating the females derived from Iso-Y lines with testosterone would indeed be an interesting follow-up study. We have now included this to the discussion on lines 501-503. We can speculate a bit further on our hypothesis of a Y-autosome epistatic interaction with previous work using the source population (Paria). Guppies from Paria were treated with testosterone by Gordon et al. 2011 (*Evolution*, **66**: 912-918) and this research found that females exhibited colour. This is intriguing given that Paria is known to have a strong Y-linked colour patterns (i.e. sons resemble their fathers) when compared to other populations (Houde 1992, *Heredity*, **69**: 229-235). If the Y chromosome was not involved in colour patterning, we would expect patrilineal inheritance to disappear after 2 generations (to account for a dominant effect of the autosome). Given that we created distinct Iso-Y lines after ~40 generations of introgression, we have strong evidence for the role of the Y in colour patterning. We have expanded on this in the discussion at lines 493-503 to more fully explain our hypothesis. We are currently testing this hypothesis more directly with a series of crosses among lines (see lines 499-501).

L. 387 (and several points throughout that form the framework on this study). Authors bring up the idea that sexual conflict is expected to move colour genes to the Y chromosome in guppies. However, the theory of sexual conflict does not actually apply to modern guppy coloration (a longstanding misunderstanding in guppy literature). Sexual conflict exists when an allele is beneficial to one sex but detrimental to the other sex. But there are two ways to resolve sexual conflict: either by moving genes to the sex-limited chromosome (as proposed here) or by sex limited gene regulation- that is to say by expressing that gene only in the sex which benefits from it. Once conflict is resolved (such as the evolution of gene regulation) then sexual conflict is no longer exerting selection to move genes to the sex-limited chromosome. In guppies – expression of coloration genes depends entirely on testosterone (as evidence by numerous studies, included citations in this the paper, that show females express colour genes when given testosterone) and thus color genes in guppies are not under sexual conflict.

We thank the reviewer for clarifying these arguments. Indeed, our results can not be explained by a sexual antagonistic loci underlying colour. Under sexual antagonism, you would predict that both females and males would be out of HWE for our candidate loci (i.e. females with the allele are selected against and males are selected for). What we find instead is that it is only males which fall outside of HWE. We hypothesise that a deleterious Y-autosome interaction could be causing this signal (lines 509-513). However, we recognise that more sampling and specific crosses are needed (see lines 525-528, 499-501). We have therefore updated the introduction (lines 49-53,56) and discussion at lines 505-507 & 567-568) to reframe the results in a Y-linkage polymorphism framework rather than a sexual conflict framework.

Several points throughout the manuscript (including abstract, introduction and discussion) the authors suggest that the field believes most of the color genes in guppies are on the Y (eg. L. 347). However, this is a misrepresentation as even the original work by Fisher showed only

components of guppy color patterns are Y inherited. To my knowledge there has never been a study of copy coloration that suggests that ALL male coloration is entirely inherited on the Y.

We agree and have updated the writing to be clearer in that there is both Y-linkage and autosomal colour patterns (lines in abstract 27-28, lines in introduction 74-86, lines in discussion 472-473). However, given that our lines show high Y-linkage, we predicted LG12 was where we would find consistent differences between the lines. The result on LG1 was unexpected and we think provides a nice example for Y-linkage without being physically linked to the Y, which could therefore be a result of Y-autosome interactions.

REVIEWER COMMENTS

Reviewer #1 (Remarks to the Author):

The authors have done a good job responding to the comments of all reviewers. In particular, I agree with their response to point 3 of reviewer 2. There is no a priori reason to expect an enrichment of color genes on LG1, even if it is associated with the color polymorphisms among males. They identify many excellent candidate genes, known to play a role in pigmentation in other systems, in their candidate regions on LG1. Mutations in these genes could quite plausibly contribute to the male color differences in guppies, even if they are not "enriched" here.

I also think they responded well to the first point of Reviewer 2. It is formally possible that they captured haplotypes segregating in the source population that are somehow maintained due to low recombination on LG1. However, it would be quite unlikely that only this chromosome would be differentiated in all 4 lines, which were selected based on color. The authors might further point out that IsoY6 and IsoY8 are the least phenotypically divergent lines, and they also show the lowest genetic differentiation in LG1 PC1. It would also be important to know whether LG1 has similar recombination rates to the rest of the genome (it is known that LG12 is lower). One would have to hypothesize much lower rates of recombination on LG1 than the rest of the genome to maintain these haplotypes over 40 generations.

I have three minor comments:

L387-389: perhaps cite Almeida et al. 2021 here.

L406: define SV here.

L531-536: I think you need to explicitly state that what the evidence is against duplication/translocations from LG1 to the Y chromosome explaining these patterns; i.e. read coverage in this part of the genome is not higher (as is expected for a gene that has duplicated and translocated elsewhere in the genome).

Additional comments in light of Reviewer 2's concerns:

I think the first concern of Reviewer 2 could be addressed easily by the authors if they provide information on the number of backcross generations they did, and then provide an estimate of how much of the genome should be homogenised, as each generation reduces the proportion of the original parental population present in the genome by half. For example, after 5 backcross generations (which seems to be fewer than those done in this manuscript but the authors will need to confirm), there should be only 3.125% percent of the original parental genome remaining. Since there are 23 guppy chromosomes, each chromosome is roughly 4% of the genome. You would then have to hypothesise that in each of the four independent lines, only chromosome 1 was not replaced by chance. The chances of this are very low unless there is really no recombination on chromosome 1. However, I looked at the Bergero et al. 2019 PNAS paper, which presents a linkage map for guppies, and there is no evidence that chromosome 1 shows unique patterns of recombination relative to the other chromosomes. Certainly, one would expect some outliers in F_{st} in the regions of low recombination in the genome, and the authors could perhaps see whether the tails of their Z- F_{st} PC1 distributions fall in the regions of low recombination on the different chromosomes and whether the percent of the genome with this pattern (outside of Chr 1 and 12) is in line with that expected based on the number of backcross generations.

I do think the authors should directly address this concern of Reviewer 2 in the manuscript, by adding these details about the number of backcross generations, the expected percent of the original parental genome remaining, and data on recombination on chromosome 1 relative to other chromosomes, which seems to be available from other work in this system.

The authors can also address the second concern of Reviewer 2 by more fully discussing this in their

manuscript. I would agree that their results of low diversity within and high divergence between iso-Y lines is expected by the breeding design. However, I was confused by the line numbers referred to by Reviewer 2, as these lines in the manuscript do not seem to correspond with the statements of the reviewer.

Reviewer #2 (Remarks to the Author):

I appreciate the time the authors have taken to respond to my comments and revise the manuscript accordingly. However, unfortunately, some of my major concerns have not been remedied.

1) I still have concerns over whether the authors can definitively link a region on Chr 1 with colour phenotypes using their breeding design. If I understand correctly, the 4 lines have been bred since 2012 with backcrossing into the stock population every 2-3 generations. A fundamental assumption is that this breeding design is sufficient to homogenise the genome of each of the 4 lines with the stock population so that only regions associated with colour will differ. However, it's not clear to me whether this is actually the case.

For instance, if genomes had been homogenised so that only regions associated with colour were retained, we would expect F_{st} between the lines to be low across the majority of the genome as they have all been backcrossed to the same stock population. Only Chr 12 would be an outlier due to the absence of Y recombination. However, we can see from Fig 1a that this is not the case. Many chromosomes show significantly elevated F_{st} (e.g. Chr 22, Chr 19). Either this suggests that all of these regions are associated with colour, which seems implausible, or that there has been insufficient backcrossing to homogenise the genome. If the latter, this weakens confidence in the results.

Therefore, it seems possible to me at the moment that different haplotypes of Chr 1 could have been retained in each line by chance simply because this region exhibits lower recombination than the rest of the genome. Studies with analogous designs in *Drosophila* typically get around this problem using many more iso-male lines, but obviously that is not feasible in this case.

In essence, how can the authors be sure that the number of backcrossing events in this design has been sufficient to completely homogenise the genome between the stock population and each line, including those regions that recombine very infrequently? I imagine the authors could estimate this with knowledge of the actual number of backcrosses and estimates of recombination rate in guppies. Without this, I don't think the authors can confidently say that Chr 1 is associated with colour.

2) Unfortunately, the authors haven't really addressed my comments regarding their finding that the Y chromosome is diverged between the iso-Y lines and shows low diversity. This is exactly what is expected from their breeding design but I don't think this is explained sufficiently clearly in the manuscript. For instance, in L177 onwards, the finding that LG12 is divergent is discussed as a result that requires further exploration, rather than a confirmation of the breeding design. This also applies to the discussion of results on L192 onwards.

L97 I don't think you need to abbreviate negative frequency dependent selection.

L249 I don't think you need to abbreviate allele frequency

Reviewer #3 (Remarks to the Author):

Paris et al's revision to the manuscript "A large and diverse autosomal haplotype is associate with sex-linked colour polymorphism in the guppy" does a very nice job of addressing my previous concerns. I especially appreciate that they cleared up some of their discussions of sexual conflict. In

doing so I feel that the true potential of this manuscript is able to come through and it offers some very interesting insights regarding the potential mechanism of color inheritance in guppies – a longstanding enigma in need of fresh ideas like those put forward in this manuscript. I have only one additional comment that I think authors could address.

L. 431-433. Regarding the wild caught individuals that were phased, care should be taken to not over interpret data. “Breakpoints were primarily located in females, except for BP3 and BP4. This difference between the sexes in the conservation of these breakpoints suggests sex-biased recombination in breaking up the overall haplotype.” However, there were only 6 BPs total, and there were twice as many females as males that were analyzed. Furthermore, if I understand correctly, the wild individuals that were sampled were adults caught in the field. Therefore, the males being sampled had already been exposed to natural selection. If this region really has an important role in coloration (as shown here) then we would expect males to not have a normal distribution of possible genetic variation in this region, but it still doesn’t mean there is sex-biased recombination. Sex-biased recombination of LG1 is an interesting idea, but I think in need of different data before it can be evaluated.

I’m looking forward to seeing your followup studies on the LG1-LG12 interaction!

~Ben Sandkam

REVIEWER COMMENTS

Reviewer #1 (Remarks to the Author):

The authors have done a good job responding to the comments of all reviewers. In particular, I agree with their response to point 3 of reviewer 2. There is no a priori reason to expect an enrichment of color genes on LG1, even if it is associated with the color polymorphisms among males. They identify many excellent candidate genes, known to play a role in pigmentation in other systems, in their candidate regions on LG1. Mutations in these genes could quite plausibly contribute to the male color differences in guppies, even if they are not "enriched" here.

I also think they responded well to the first point of Reviewer 2. It is formally possible that they captured haplotypes segregating in the source population that are somehow maintained due to low recombination on LG1. However, it would be quite unlikely that only this chromosome would be differentiated in all 4 lines, which were selected based on color. The authors might further point out that IsoY6 and IsoY8 are the least phenotypically divergent lines, and they also show the lowest genetic differentiation in LG1 PC1. It would also be important to know whether LG1 has similar recombination rates to the rest of the genome (it is known that LG12 is lower). One would have to hypothesize much lower rates of recombination on LG1 than the rest of the genome to maintain these haplotypes over 40 generations.

Thank you for your comments arbitrating the concerns of reviewer 2, please see further detailed responses below.

I have three minor comments:

L387-389: perhaps cite Almeida et al. 2021 here.

Done (now on line 390):

"In contrast to LG1, which showed that elevated intersex \$F_{ST}\$ was driven by reduced male diversity, the high diversity observed here is consistent with a hypothesis of multiple diverse Y-haplotypes and NFDS on the Y (Almeida et al 2021)"

L406: define SV here.

Done (now on lines 407-408):

"Structural variant (SV) analyses using short-read and long-read data did not show support for SVs in Region 2 nor Region 2-NP"

L531-536: I think you need to explicitly state that what the evidence is against duplication/translocations from LG1 to the Y chromosome explaining these patterns; i.e. read coverage in this part of the genome is not higher (as is expected for a gene that has duplicated and translocated elsewhere in the genome).

We have added this information to lines 539-541:

"Read coverage was not increased in the LG1 region, and we found no evidence of reduced mapping quality, which would be expected if this region was the result of a duplication or translocation event"

Additional comments in light of Reviewer 2's concerns:

I think the first concern of Reviewer 2 could be addressed easily by the authors if they provide information on the number of backcross generations they did, and then provide an estimate of how much of the genome should be homogenised, as each generation reduces the proportion of the original parental population present in the genome by half. For example, after 5 backcross generations (which seems to be fewer than those done in this manuscript but the authors will need to confirm), there should be only 3.125% percent of the original parental genome remaining. Since there are 23 guppy chromosomes, each chromosome is roughly 4% of the genome. You would then have to hypothesise that in each of the four independent lines, only chromosome 1 was not replaced by chance. The chances of this are very low unless there is really no recombination on chromosome 1. However, I looked at the Bergero et al. 2019 PNAS paper, which presents a linkage map for guppies, and there is no evidence that chromosome 1 shows unique patterns of recombination relative to the other chromosomes. Certainly, one would expect some outliers in F_{st} in the regions of low recombination in the genome, and the authors could perhaps see whether the tails of their Z- F_{st} PC1 distributions fall in the regions of low recombination on the different chromosomes and whether the percent of the genome with this pattern (outside of Chr 1 and 12) is in line with that expected based on the number of backcross generations.

I do think the authors should directly address this concern of Reviewer 2 in the manuscript, by adding these details about the number of backcross generations, the expected percent of the original parental genome remaining, and data on recombination on chromosome 1 relative to other chromosomes, which seems to be available from other work in this system.

The authors can also address the second concern of Reviewer 2 by more fully discussing this in their manuscript. I would agree that their results of low diversity within and high divergence between iso-Y lines is expected by the breeding design. However, I was confused by the line numbers referred to by Reviewer 2, as these lines in the manuscript do not seem to correspond with the statements of the reviewer.

We thank Reviewer 1 for these constructive suggestions for how to respond to the concerns raised by Reviewer 2. We have, in general, followed these recommendations. The only recommendation that we have been unable to perform is quantitatively assessing whether the tails of the Z- F_{st} PC1 distributions from other chromosomes fall within known regions of low recombination. Unfortunately, the available linkage maps are too sparse to identify definitive low recombination regions and perform a quantitative assessment. As an alternative, we plot smoothed recombination against Z- F_{st} PC1 for chromosomes 19 and 22. On LG22, we see a qualitative association with low recombination. On LG19, we explore pairwise F_{ST} and do not see the same consistent differentiation as observed on LG1. Further detail can be found below in our responses to Reviewer 2.

As suggested above, we now point out in the manuscript that indeed, Iso-Y6 and Iso-Y8 show the lowest divergence on LG1 (lines 489-491):

“Moreover, we found that Iso-Y6 and Iso-Y8 are the least phenotypically divergent lines, and they also showed the lowest genetic differentiation on LG1”.

We have also added an additional supplementary figure (Supplementary Fig. 7) which shows Z- F_{ST} -PC1 together with a smoothed recombination landscape on LG1 (from Whiting et al 2021), showing no association between divergence and reduced recombination. We have added text to the manuscript in the results (lines 190-191):

“We found no relationship between $Z-F_{ST}$ PC1 scores and the recombination landscape on LG1 (Supplementary Fig. 7).”

And to the discussion (lines 486-487):

“However, we think this scenario is unlikely; there is no evidence of unusually low recombination on LG1^{30,33} (Supplementary Figure 7)”

We describe in full how we have implemented these revisions in the detailed responses to Reviewer 2, below.

Reviewer #2 (Remarks to the Author):

I appreciate the time the authors have taken to respond to my comments and revise the manuscript accordingly. However, unfortunately, some of my major concerns have not been remedied.

1) I still have concerns over whether the authors can definitively link a region on Chr 1 with colour phenotypes using their breeding design. If I understand correctly, the 4 lines have been bred since 2012 with backcrossing into the stock population every 2-3 generations. A fundamental assumption is that this breeding design is sufficient to homogenise the genome of each of the 4 lines with the stock population so that only regions associated with colour will differ. However, it's not clear to me whether this is actually the case.

For instance, if genomes had been homogenised so that only regions associated with colour were retained, we would expect F_{ST} between the lines to be low across the majority of the genome as they have all been backcrossed to the same stock population. Only Chr 12 would be an outlier due to the absence of Y recombination. However, we can see from Fig 1a that this is not the case. Many chromosomes show significantly elevated F_{ST} (e.g. Chr 22, Chr 19). Either this suggests that all of these regions are associated with colour, which seems implausible, or that there has been insufficient backcrossing to homogenise the genome. If the latter, this weakens confidence in the results.

Therefore, it seems possible to me at the moment that different haplotypes of Chr 1 could have been retained in each line by chance simply because this region exhibits lower recombination than the rest of the genome. Studies with analogous designs in *Drosophila* typically get around this problem using many more iso-male lines, but obviously that is not feasible in this case.

In essence, how can the authors be sure that the number of backcrossing events in this design has been sufficient to completely homogenise the genome between the stock population and each line, including those regions that recombine very infrequently? I imagine the authors could estimate this with knowledge of the actual number of backcrosses and estimates of recombination rate in guppies. Without this, I don't think the authors can confidently say that Chr 1 is associated with colour.

Thank you for your comments. We agree with Reviewer 1 in that we would have been extremely unlucky to have detected a spurious relationship between the LG1 candidate region and colour differences in lines, as each generation reduces the proportion of the original parental population present in the genome by half (Hill 1998 Genetics, reviewed in Hospital 2005 Phil Trans R Soc

London B). Indeed, the standard number of backcrossing generations is 10 (i.e. 99.99% of the stock population genome has been introgressed). Here, we perform 13 to 20 backcrossed generations (every 2-3 generations for 40 generations). This has been added to the introduction at lines 107-109:

“Given that each backcrossed generation theoretically reduces the parental genome by half, we expect more than 99.99% of the genome to be homogenised through this approach (Hill, 1998)”

There is no evidence of reduced recombination on LG1 compared to other chromosomes (Bergero et al 2019 PNAS; Whiting et al 2021 bioRxiv:10.1101/2021.03.18.435980). We have added a supplementary figure (Figure S7) showing the Z-FST-PC1 together with smoothed recombination rates for LG1. We would also like to emphasise, as pointed out by Reviewer 1, that Iso-Y6 and Iso-Y8 are the least phenotypically divergent, and show the lowest differentiation on LG1.

To fully address this, we have added a paragraph to the discussion on lines 484-491:

“An alternative explanation for the association between LG1 Region-2 haplotype and colour is unusually low recombination on LG1 and insufficient backcrossing, resulting in a spurious relationship. However, we think this scenario is unlikely; there is no evidence of unusually low recombination on LG1^{30,33} (Supplementary Figure 7), and as each generation reduces the proportion of parental genome by half, just <0.01% of the parental genome should remain after 13-20 generations of backcrossing. Moreover, we found that Iso-Y6 and Iso-Y8 are the least phenotypically divergent lines, and they also showed the lowest genetic differentiation on LG1.”

We would also like to point out that by Z-score, we mean an upper 95% quantile on the scaled Z-scores of the PCA. To limit any potential confusion, we have changed the text to reflect that we are discussing a quantile, not a true Z-score related to standard deviations from the mean. Thus, 3 represents the upper 95% quantile and so any SNPs which show a score above this value do not necessarily indicate “significantly elevated FST”. Moreover, we do not need to rely only on this quantile threshold because we go further by looking directly at the allele frequency differences among the lines (Figure 1c) and pairwise FST (Figure S5); we can see clear and consistent differences at a specific haplotype on LG1.

We do not agree with the reviewer that because LG22 and LG19 show increased Z-FST scores that our breeding design hasn't homogenised the genome. Both LG22 and LG19 have Z-FST scores above 3. However, LG1 and LG12 are the only chromosomes to have Z-FST scores above 6 (Figure 1a). Secondly, in Figure S4 we show that the LG1 and LG12 have the highest percentage of SNPs with Z-FST PC1 score above 3 (the upper 95% quantile): 23.5% and 29% respectively. LG22 and LG19 have 6% and 5%, respectively (detailed on lines 172-177). Moreover, the mean Z-FST PC1 scores for LG1 and LG12 are 1.48 and 1.62, respectively. For LG19 and LG22, the mean Z-FST scores are 0.2 and 0.78, respectively.

Whilst we cannot quantitatively assess whether or not the tails of high Z-FST PC1 distributions fall within known regions of low recombination on LG19 and LG22 (due to the sparseness of available linkage maps), we can look at Z-FST PC1 and can compare them to their smoothed recombination landscapes (extracted from Whiting et al 2021):

Firstly, we see no association with Z-FST-PC1 SNPs and low recombination on LG1 (now detailed in the manuscript in Figure S7). Secondly, we also see that on these other chromosomes, the PC loadings do not load as positively as they do on LG1, and we also do not see clearly defined regions as in LG1. Then, on LG22, we see that indeed Z-FST PC1 SNPs appear to associate with a region of low recombination. We can also look at the raw pairwise FST differences, where we see there is no consistent differentiation between the Iso-Y lines on LG19 and LG22:

We would be happy to add more of this additional information to the supplement based on the editor's recommendation, but we note that the supplement already contains 24 figures.

2) Unfortunately, the authors haven't really addressed my comments regarding their finding that the Y chromosome is diverged between the iso-Y lines and shows low diversity. This is exactly what is expected from their breeding design but I don't think this is explained sufficiently clearly in the manuscript. For instance, in L177 onwards, the finding that LG12 is divergent is discussed as

a result that requires further exploration, rather than a confirmation of the breeding design. This also applies to the discussion of results on L192 onwards.

We agree that yes, our finding that LG12 is divergent among the Iso-Y lines is what we expected to see, but the patterns of differentiation were not consistent with either the whole Y, nor a specific part of the Y being consistently divergent. Unfortunately, as stated above by Reviewer 1, these line numbers don't match in the manuscript, so it's difficult to directly address the precise sections referred to, but we have added additional information on our expectations of Y chromosome divergence in the discussion on lines 472-482 to make this clearer:

“We had hypothesised that the Y-linked colour pattern genes in our Iso-Y lines would be located on the sex chromosome, LG12. If colour pattern traits are fully Y-linked, and each Iso-Y line's unique Y haplotype is fully inherited through the patriline, we predict a consistent difference in a region along LG12. Instead, we see that the entire chromosome shows moderate divergence, as predicted by the breeding design, but with no clear consistently differentiated region. All guppy chromosomes are acrocentric and show evidence of male heterochiasmy; yet recombination events are particularly rare on LG12, implying that LG12 may decompose at a slower rate compared to autosomes. Our LG1 haplotype on the other hand, shows consistent differentiation between the lines. The near fixation of the LG1 Region-2 haplotype between the lines is strong evidence for this region harbouring genes involved in colour”

L97 I don't think you need to abbreviate negative frequency dependent selection.

Negative frequency dependent selection appears 6 times in the manuscript, and NFDS is a commonly used abbreviation. We would therefore like to keep it.

L249 I don't think you need to abbreviate allele frequency

As above, allele frequency occurs 29 times in the manuscript, and so we would like to keep AF.

Reviewer #3 (Remarks to the Author):

Paris et al's revision to the manuscript “A large and diverse autosomal haplotype is associated with sex-linked colour polymorphism in the guppy” does a very nice job of addressing my previous concerns. I especially appreciate that they cleared up some of their discussions of sexual conflict. In doing so I feel that the true potential of this manuscript is able to come through and it offers some very interesting insights regarding the potential mechanism of color inheritance in guppies – a longstanding enigma in need of fresh ideas like those put forward in this manuscript. I have only one additional comment that I think authors could address.

L. 431-433. Regarding the wild caught individuals that were phased, care should be taken to not over interpret data. “Breakpoints were primarily located in females, except for BP3 and BP4. This difference between the sexes in the conservation of these breakpoints suggests sex-biased recombination in breaking up the overall haplotype.” However, there were only 6 BPs total, and there were twice as many females as males that were analyzed. Furthermore, if I understand correctly, the wild individuals that were sampled were adults caught in the field. Therefore, the males being sampled had already been exposed to natural selection. If this region really has an important role in coloration (as shown here) then we would expect males to not have a normal distribution of possible genetic variation in this region, but it still doesn't mean there is sex-biased

recombination. Sex-biased recombination of LG1 is an interesting idea, but I think in need of different data before it can be evaluated.

Thank you for this comment; this is a good point. We have removed this sentence from the manuscript.

I'm looking forward to seeing your followup studies on the LG1-LG12 interaction!

Thanks - us too!

~Ben Sandkam

REVIEWER COMMENTS

Reviewer #2 (Remarks to the Author):

I thank the authors for their thoughtful and detailed response. I am also pleased to see that reviewer 1 agrees with the importance of showing that the Iso-Y lines have been sufficiently backcrossed into the parental population so that the genome is fully homogenized across lines. I think the strength of the results hinge on this.

I completely understand the authors' argument that LG1 is more divergent than the other LG. But, as the authors show in their response, other LGs also have differentiated SNPs. I wouldn't expect this if the lines were truly homogenized with the parental genome, and I can't find an explanation for why this pattern exists in their data.

Specifically, I understand the authors' argument that allele frequency differences among lines and pairwise FST are consistently different on LG1.

'Moreover, we do not need to rely only on this quantile threshold because we go further by looking directly at the allele frequency differences among the lines (Figure 1c) and pairwise FST (Figure S5); we can see clear and consistent differences at a specific haplotype on LG1.'

However, I don't find figures (1c, S5) very convincing as they don't show any other linkage groups apart from LG1. If the genomes had truly been homogenized across lines, then pairwise FST and allele frequency differences should be 0 across other LG - but the authors haven't showed this.

Again, I understand the argument that the Z-FST PCI is definitely higher for LG1 than the other LG, but this is just a question of scale and seems rather subjective.

I would be happy to recommend the paper for publication if the authors can show that allele frequency differences among lines and pairwise FST are ~ 0 on other linkage groups. This will indicate the lines are homogenized with the parental genome. If not, the authors need to explain clearly why this isn't a problem.

Reviewer #2 (Remarks to the Author):

I thank the authors for their thoughtful and detailed response. I am also pleased to see that reviewer 1 agrees with the importance of showing that the Iso-Y lines have been sufficiently backcrossed into the parental population so that the genome is fully homogenized across lines. I think the strength of the results hinge on this.

Below we present evidence on the following points:

- We expect there to be some high-scoring SNPs between pairwise Iso-Y line comparisons on other autosomes due to chance. Using calculated empirical p-values, only LG1 and LG12 show many more of these than expected by chance alone.
- The expectation that $F_{ST} \approx 0$ as an indication of true homogenisation is incorrect for two reasons:
 1. We expect localised regions of pairwise differentiation between any particular Iso-Y lines due to our breeding design (i.e., individual outliers are expected based on chance, drift and bottlenecking).
 2. We do not expect F_{ST} to be 0, as it is zero-bounded with a right-tailed distribution due, in part, to sampling variance.
- There is no association between highly differentiated SNPs on other autosomes and recombination rate coldspots using available linkage maps (from other guppy populations in Northern Trinidad)
- There is no association between highly differentiated SNPs on other autosomes and recombination rate coldspots using lostruct local PCA MDS1 outliers from our natural data (used as an indication of potential low recombination regions from the same guppy population as our Iso-Y lines)
- Crucially, none of these aforementioned reasons can explain our *consistent* differences on LG1.

I completely understand the authors' argument that LG1 is more divergent than the other LG. But, as the authors show in their response, other LGs also have differentiated SNPs. I wouldn't expect this if the lines were truly homogenized with the parental genome, and I can't find an explanation for why this pattern exists in their data.

Specifically, I understand the authors' argument that allele frequency differences among lines and pairwise F_{ST} are consistently different on LG1.

'Moreover, we do not need to rely only on this quantile threshold because we go further by looking directly at the allele frequency differences among the lines (Figure 1c) and pairwise F_{ST} (Figure S5); we can see clear and consistent differences at a specific haplotype on LG1.'

However, I don't find figures (1c, S5) very convincing as they don't show any other linkage groups apart from LG1. If the genomes had truly been homogenized across lines, then pairwise F_{ST} and allele frequency differences should be 0 across other LG - but the authors haven't showed this.

Again, I understand the argument that the Z- F_{ST} PCI is definitely higher for LG1 than the other LG, but this is just a question of scale and seems rather subjective.

Other LGs have differentiated SNPs because under randomness all chromosomes are expected to have 5% of their SNPs in the upper 5% tail; that is, there will always be some very small p-

values by chance, and thus some differentiated SNPs are expected on all chromosomes. To demonstrate this, we calculated the empirical p-value for each SNP's Z-FST-PC1 score with respect to all other SNPs in the genome. If there are no true positives, then the p-values (for any given chromosome) should be more or less uniformly distributed between 0 and 1. If we draw the distribution of empirical p-values per chromosome, we can see this effect very clearly: only LG1 and LG12 have many more of these SNPs than one would expect by chance:

Distribution of empirical p-values calculated from Z-FST-PC1 scores for each chromosome. Vertical red line marks 5% cut-off

I would be happy to recommend the paper for publication if the authors can show that allele frequency differences among lines and pairwise F_{ST} are ~ 0 on other linkage groups. This will indicate the lines are homogenized with the parental genome. If not, the authors need to explain clearly why this isn't a problem.

How close to ~ 0 should F_{ST} be to show true homogenisation? Most chromosomes show mean pairwise $F_{ST} < 0.1$ (Supplementary Table 5):

	Iso-Y10 vs Iso-Y6	Iso-Y10 vs Iso-Y8	Iso-Y10 vs Iso-Y9	Iso-Y6 vs Iso-Y8	Iso-Y6 vs Iso-Y9	Iso-Y8 vs Iso-Y9
LG1	0.11	0.11	0.09	0.05	0.26	0.24
LG2	0.07	0.06	0.04	0.11	0.09	0.03
LG3	0.05	0.09	0.06	0.12	0.05	0.13
LG4	0.07	0.09	0.05	0.11	0.10	0.08

LG5	0.05	0.06	0.04	0.05	0.09	0.10
LG6	0.09	0.03	0.06	0.07	0.11	0.09
LG7	0.13	0.07	0.08	0.07	0.07	0.06
LG8	0.09	0.06	0.05	0.07	0.15	0.15
LG9	0.06	0.06	0.04	0.09	0.03	0.08
LG10	0.09	0.04	0.04	0.08	0.05	0.04
LG11	0.08	0.04	0.06	0.07	0.16	0.10
LG12	0.12	0.04	0.21	0.13	0.22	0.20
LG13	0.05	0.05	0.03	0.07	0.06	0.08
LG14	0.09	0.05	0.08	0.05	0.16	0.13
LG15	0.05	0.04	0.09	0.08	0.06	0.10
LG16	0.06	0.06	0.03	0.03	0.06	0.06
LG17	0.06	0.03	0.03	0.04	0.08	0.04
LG18	0.08	0.07	0.03	0.08	0.08	0.09
LG19	0.11	0.02	0.06	0.11	0.16	0.08
LG20	0.07	0.05	0.09	0.07	0.10	0.07
LG21	0.14	0.07	0.04	0.09	0.12	0.09
LG22	0.12	0.11	0.08	0.19	0.08	0.14
LG23	0.08	0.05	0.04	0.06	0.09	0.07

Moreover, we disagree with the reviewer that $F_{ST}=0$ is indicative of full homogenisation. Firstly, in light of our breeding design, consider that each of these Iso-Y lines were founded from a natural population. Each line was established by crossing one male colour with a few females from the original natural population. In subsequent generations, 2-3 males from each line were crossed with 4-6 females from the same line or, every 2-3 generations, from the stock population. These females would have variation both at colour-specific genes and at regions unrelated to colour, as part of the segregating variation within a population. Because of the small number of individuals in this breeding design, this variation was submitted to strong bottlenecks, which means that genetic drift will also have an important influence in driving random changes in allele frequencies. This will drive areas of “high” differentiation between a particular pair of Iso-Y lines, which is highly unlikely to be present in other Iso-Y line comparisons. Therefore, isolated outliers are to be expected. Note however, that this bottleneck effect should not lead to a *consistent* region of differentiation among all Iso-Y lines, like we see on LG1, since allele frequency change should be random with respect to genome regions within each line.

Next, it should be noted that the Iso-Y lines were sequenced using pool-sequencing, so our expectation is that the allele frequency differences (AFD) (and F_{ST}) between each Iso-Y line will be non-0 because they are zero-bound, and will have a right-tailed distribution with a standard deviation reflecting sampling variance. Using mathematical derivations from Gautier et al 2013 (Mol. Ecol. **22**, 14: 3766-3779), we can estimate our sampling variance (using allele frequencies expected to be present in each pool without selection):

Note that this sampling variance is generated by the sampling of DNA molecules within a pool-sequencing experiment, which is dependent on read depth (here 120x) and the number of individuals sequenced (here 48). These values do not include additional variance generated by the breeding design (which depend on the effective population size and number of generations).

Using this curve, we can then actually estimate what effect this sampling variance has on our empirical estimates of AFD and FST. We have performed an assessment of this by analysing the sampling variance on the allele frequencies *within* each of the Iso-Y lines, finding that this variance produces a median AFD of 0.07 and a median FST of 0.01, with maximums of 0.4 and 0.45, respectively. The plots below show this in more detail.

This demonstrates that high AFD/FST can happen by chance. Given these estimates are derived *within* a pool, they are likely conservative underestimates of the effects of sampling variance on FST *among* pools. We wish to also emphasise that the variance shown in these graphs is just

one source of variance in AFD and pairwise F_{ST} , so these values actually underestimate those we should see under purely neutral processes in the pool-seq data. Thus, both the sampling variance generated by pool-sequencing and genetic drift during the breeding design have non-negligible effects, elevating AFD and F_{ST} between pairs of Iso-Y lines above 0. However, we can be confident that LG1 Region 2 is not a false positive because the differentiation occurs *consistently* in all four Iso-Y lines.

Let's now assume that the Iso-Y lines are not homogenised. A lack of homogenisation would suggest a genome-wide effect of recombination on differentiation, where low recombination regions tend to be more differentiated. To this end, we have now performed a quantitative analysis of recombination and F_{ST} , by reassessing all available recombination data.

Quantitative assessment of recombination and F_{ST} differentiation across the genome. a) Heatmap showing correlations (Kendall's tau) between the Iso-Y line pairwise F_{ST} estimates and recombination. Black represents a negative correlation (F_{ST} associated with low recombination) and white represents a positive correlation (F_{ST} associated with high recombination). b) boxplots displaying the chromosome-wide correlation (Kendall's tau) between recombination and mean F_{ST} (in 100 Kbp windows). c) histogram of the distribution of genome-wide F_{ST} -recombination correlation.

Although available guppy linkage maps are sparse, they capture the broad acrocentric nature of recombination in the guppy chromosomes (see Supplementary Fig. 5) (Lisachov AP., *et al.*, 2015, *Zebrafish*. **12**, 174–180; Charlesworth D., *et al.*, 2020, *Mol. Biol. Evol.* (2020), doi:10.1093/molbev/msaa187; Charlesworth D., *et al.*, 2020, *G3: Genes, Genomes, Genetics*. **10**, 3639–3649). This sets up the prediction that if the Iso-Y lines are not fully homogenised, the ends of chromosomes closest to the acrocentric telomere should be the least homogenised, and thus the most differentiated. To thoroughly test this, we have performed an analysis of the correlation between recombination from these genetic linkage maps and pairwise FST (Supplementary Fig. 6). This clearly shows that there are no correlations with recombination on LG1, and correlations with recombination are variable across pairs and in most cases, centred around zero (panel B). This also rules out a genome-wide signature of recombination on differentiation. We have also added the pairwise FST for the other LGs showing high-scoring Z-FST PC1 SNPs, along with the smoothed recombination maps of each chromosome (Supplementary Fig. 5).

But just to be certain, we can go even further by assuming that these analyses can fail to capture local regions of low recombination (coldspots) specific to our focal populations (Northern drainage populations, not used in genetic linkage mapping studies). We can overcome this by using lostruct local PCA (Li H. & Ralph P, 2019, *Genetics*. **211**, 289–304) on the natural data. Mean relatedness is expected to be lower between individuals in recombination coldspots, and thus we can harness the signatures of outlier MDS regions (as a proxy for the reductions in this relatedness) using this approach. In this method, inversions, linked selection and recombination coldspots are the greatest contributors towards deviations in MDS space, all of which would be predicted to reduce homogenisation among the Iso-Y lines. Again, this sets up the prediction that these regions should be associated with differentiation among the lines if they reduce homogenisation and the lines are not fully homogenised.

To this end, we assessed these relationships across the whole genome, by exploring the association between lostruct local PCA outliers on MDS1 and Z-FST PC1. We have coloured regions of chromosomes which harbour high-scoring Z-FST PC1 SNPs (LG1, LG3, LG8, LG12, LG19, LG21, LG22).

Assessment of correlation between Z-FST PC1 and MDS1 outliers from lostruct. Points are coloured by chromosome (LG1 - orange; LG3 - light blue; LG8 - yellow; LG12 - green; LG19 - dark blue; LG21 - red; LG22 - pink). Only LG1 and LG12 show high Z-FST PC1 scores and significantly elevated MDS1 scores.

We find that there is no association between MDS1 and FST differentiation, except in LG1, meaning that overall these 'outliers' are likely not due to differences in local recombination rates or inversions. Overall, this demonstrates that it is only some regions of LG1 (and Iso-Y9 specific regions of LG12) which show an association between Z-FST PC1 and MDS1, and also that MDS1 signatures like LG1 do not automatically produce elevated differentiation among the Iso-Y lines because of low recombination (or experimental design). As such, there is no reason to assume *a priori* that low recombination regions should produce elevated differentiation, as would be expected if the Iso-Y lines were not fully homogenised. In summary, these quantitative analyses clearly show that there are no associations with recombination and consistent differentiation.

In conclusion, the premise that our breeding design has not resulted in true homogenisation based only on evidence that FST should be ~ 0 on other autosomes is mistaken. And then, even if we assume the Iso-Y lines are not homogenised and test the signal that differentiation should relate to recombination, we also do not find evidence to support the claim that Iso-Y lines are not homogenised. Thus, our evidence that the LG1 candidate haplotype is associated with colour differences between the Iso-Y lines is strong.

REVIEWER COMMENTS

Reviewer #1 (Remarks to the Author):

Again, I think the authors have done an excellent job of responding to the criticisms of Reviewer 2. They have used multiple lines of evidence to show that these lines have been homogenized, that recombination patterns do not influence their results, and that the patterns of differentiation between the lines on LG1 are quite exceptional from the rest of the genome. This is a very rigorous study, and it makes an important contribution to the literature on sex chromosome evolution and negative frequency dependent selection.

I noted three small typos:

L51: "it's" should be "its"

L347: "changes gene density" should be "changes in gene density"

L447: "to the beginning" should be "from the beginning"

Reviewer #2 (Remarks to the Author):

I thank the authors for their thoughtful and detailed response. They have clearly put a lot of thought and time into these reanalyses and I appreciate this. Unfortunately though, they raise more questions than they answer. Comments below:

a. In my last review I suggested that the authors amend Fig S5 to show other linkage groups. This was because the authors argue that there are consistent differences on LG1 across lines but given the other LG weren't shown it was very difficult to see the extent to which this was unusual. I am pleased to see that the authors have now done this.

However, I disagree with the authors statement that this new figure shows 'inconsistent regions of localized differentiation across other LG'. If you look, there are many peaks that overlap. E.g in LG3 there is a clear peak at the end of the chromosome that is consistent across 3/6 of the lines. In LG22 there is a clear peak in the middle that is present in 4/6 lines.

In essence, I can't see how the patterns for LG1 are much different for these other LG. In theory, this could be the result of cold spots in the population.

It is unfortunately that the authors did not explain this discrepancy in their response.

b. New figure showing the empirical p-value for each SNP's Z-FST-PC1 score with respect to all other SNPs in the genome.

This figure doesn't provide extra information. It is just a different way of plotting the data in Fig 2A. LG 12 and 1 show the highest Fst values and so we would a priori expect the top 5% of SNPs to fall on these chromosomes.

The SNPs are also not independent due to linkage. Therefore, these results are entirely confounded by the size of the differentiated region.

c. Most chromosomes show mean pairwise FST < 0.1

Again, if there are cold and hot spots across each chromosome, as we know there are in guppies, we would expect low pairwise FST across the entire chromosome.

d. Supplementary Fig 6.

There is insufficient information on these analyses. Where does the recombination data come from? The strength of these results rests on the resolution and accuracy of the recombination data. As the

authors themselves acknowledge, guppy linkage maps are sparse. Is that data from Whiting et al 2021 (bioRxiv 2021.03.18.435980)? If so, they only conducted four F2 full-sib intercrosses, and so their recombination estimates will be very gross estimates.

e. New figure 'We find that there is no association between MDS1 and FST differentiation'

I'm not familiar with the lostruct local PCA method. I've now read the paper but I don't understand how a small region can simultaneously be closely related between individuals and exhibit high sequence differentiation?

Regardless, unfortunately I think this figure weakens the authors argument. As the other reviewers pointed out, the pattern of elevated FST on LG12 is highly likely to be due to infrequent recombination arising from sex-linkage. However, it shows the same pattern in this new figure as LG1 - suggesting that reduced recombination on LG1 could be responsible for the pattern.

f. A general point about the paper. The finding that LG12 exhibits elevated FST is still discussed as though this was an unexpected result and not a natural consequence of sex-linkage.

REVIEWERS' COMMENTS

Reviewer #1 (Remarks to the Author):

Again, I think the authors have done an excellent job of responding to the criticisms of Reviewer 2. They have used multiple lines of evidence to show that these lines have been homogenized, that recombination patterns do not influence their results, and that the patterns of differentiation between the lines on LG1 are quite exceptional from the rest of the genome. This is a very rigorous study, and it makes an important contribution to the literature on sex chromosome evolution and negative frequency dependent selection.

I noted three small typos:

L51: "it's" should be "its"

L347: "changes gene density" should be "changes in gene density"

L447: "to the beginning" should be "from the beginning"

We thank reviewer 1 for their encouraging comments. All typos have now been amended in the revised manuscript.

Reviewer #2 (Remarks to the Author):

I thank the authors for their thoughtful and detailed response. They have clearly put a lot of thought and time into these reanalyzes and I appreciate this. Unfortunately though, they raise more questions than they answer. Comments below:

a. In my last review I suggested that the authors amend Fig S5 to show other linkage groups. This was because the authors argue that there are consistent differences on LG1 across lines but given the other LG weren't shown it was very difficult to see the extent to which this was unusual. I am pleased to see that the authors have now done this.

However, I disagree with the authors statement that this new figure shows 'inconsistent regions of localized differentiation across other LG'. If you look, there are many peaks that overlap. E.g in LG3 there is a clear peak at the end of the chromosome that is consistent across 3/6 of the lines. In LG22 there is a clear peak in the middle that is present in 4/6 lines. In essence, I can't see how the patterns for LG1 are much different for these other LG. In theory, this could be the result of cold spots in the population.

It is unfortunately that the authors did not explain this discrepancy in their response.

As explained in the previous revision, we expect localised regions of pairwise differentiation between any particular Iso-Y lines due to our breeding design. The peaks which overlap are not consistent between comparisons. LG3 shows a peak of differentiation unique to one Iso-Y line (Iso-Y6). LG22 shows a peak of differentiation which overlaps with a suspected recombination coldspot.

b. New figure showing the empirical p-value for each SNP's Z-FST-PC1 score with respect to all other SNPs in the genome.

This figure doesn't provide extra information. It is just a different way of plotting the data in Fig 2A. LG 12 and 1 show the highest F_{st} values and so we would a priori expect the top 5% of SNPs to fall on these chromosomes.

The SNPs are also not independent due to linkage. Therefore, these results are entirely confounded by the size of the differentiated region.

We politely disagree, and wish to keep this figure to demonstrate that LG1 and LG12 show more high-scoring SNPs than expected by chance, as explained in our previous revision.

c. Most chromosomes show mean pairwise $F_{ST} < 0.1$

Again, if there are cold and hot spots across each chromosome, as we know there are in guppies, we would expect low pairwise F_{ST} across the entire chromosome.

We showed this information in our last revision to emphasise the point that other chromosomes show low overall F_{ST} , as per the previous request that F_{ST} should be close to 0. We discuss recombination in our last revision and throughout the manuscript.

d. Supplementary Fig 6.

There is insufficient information on these analyses. Where does the recombination data come from? The strength of these results rests on the resolution and accuracy of the recombination data. As the authors themselves acknowledge, guppy linkage maps are sparse. Is that data from Whiting et al 2021 (bioRxiv 2021.03.18.435980)? If so, they only conducted four F2 full-sib intercrossovers, and so their recombination estimates will be very gross estimates.

We have amended the caption of Supplementary Figure 6, including acknowledgement that the data are broad estimates of the recombination rate in guppies.

“Recombination data were derived from four independent F2 QTL crosses as outlined in Whiting et al 2021. The linkage map consists of 6,765 markers and a length of 1,673.8 cM. To extract information on recombination rate, maps were smoothed using the smooth.spline function in R. Source data for these maps are provided as a Source Data file. Smoothing was necessary to estimate recombination over regions where individual markers were not syntenic with the published reference genome (micro-rearrangements). Larger rearrangements (such as assembly errors and inversions), where multiple markers were out of order, were manually re-oriented resulting in smoothed maps. Smoothed maps have an average of 242 markers per chromosome. These recombination data are sparse, and so likely underestimate recombination coldspots/hotspots. However, the maps represent the highest-density markers available for the species. Moreover, they capture the broad acrocentric nature of recombination in the guppy chromosomes (Lisachov et al 2015; Charlesworth et al 2020).”

e. New figure 'We find that there is no association between MDS1 and F_{ST} differentiation'

I'm not familiar with the lostruct local PCA method. I've now read the paper but I don't understand how a small region can simultaneously be closely related between individuals and exhibit high sequence differentiation?

Regardless, unfortunately I think this figure weakens the authors argument. As the other reviewers pointed out, the pattern of elevated F_{ST} on LG12 is highly likely to be due to infrequent recombination arising from sex-linkage. However, it shows the same pattern in this new figure as LG1 - suggesting that reduced recombination on LG1 could be responsible for the pattern.

We have extensively looked for the confounding effects of recombination on LG1 and found none. Reviewer 1 agrees that we have demonstrated multiple lines of evidence that recombination does not affect our results. Please see our detailed explanation of the lostruct PCA method in the previous revision where we note that outliers in MDS space from this method can be the result of 1) linked selection; 2) recombination and 3) inversions (so not just cold spots in recombination).

f. A general point about the paper. The finding that LG12 exhibits elevated F_{ST} is still discussed as though this was an unexpected result and not a natural consequence of sex-linkage.

Unfortunately, the reviewer does not provide any specific places in the manuscript which need amending. We agree that our LG12 findings (elevated F_{ST}) were expected with this breeding design. We see only one place in the discussion where we discuss the LG12 results, which we have amended so that our predictions are clearer (lines 404-411):

“If colour pattern traits are fully Y-linked, and each Iso-Y line’s unique Y haplotype is fully inherited through the patriline, then with this breeding design we would predict a consistent pattern of high differentiation in the Y-linked region on LG12 between all Iso-Y line comparisons. Instead, we see that overall, the chromosome shows moderate divergence, driven by elevated differentiation in comparisons to Iso-Y9 and localised regions of differentiation in comparisons with Iso-Y6. Whilst elevated F_{ST} across the chromosome is expected due to sex linkage, we found no evidence of a consistently differentiated region on LG12.”